# Tipping the ENSO into a permanent El Niño can trigger state transitions in global terrestrial ecosystems

Mateo Duque-Villegas[1], Juan F. Salazar[1], and Angela M. Rendón[1]

[1]GIGA, Escuela Ambiental, Facultad de Ingeniería, Universidad de Antioquia, Medellín, Colombia

**Correspondence:** Juan F. Salazar (juan.salazar@udea.edu.co)

**Abstract.** Some large-scale components of the Earth's climate system have been identified as policy-relevant "tipping elements", meaning that anthropogenic forcing and perturbations may push them across a tipping point threshold, with potential global scale impact on ecosystems and concomitant environmental and social phenomena. A pronounced change in the amplitude and/or frequency of the El Niño–Southern Oscillation (ENSO) is among such tipping elements. Here we use the Planet Simulator (PlaSim), an Earth system model of intermediate complexity, to investigate the potential impact on global climate and terrestrial ecosystems of shifting the current dynamics of the ENSO into a permanent El Niño. When forced with sea surface temperature (SST) derived from observations, the PlaSim model yields a realistic representation of large-scale climatological patterns, including realistic estimates of the global energy and water balances, and gross primary productivity (GPP). In a permanent El Niño state, we found significant differences in the global distribution of water and energy fluxes, and associated impacts on GPP, indicating that vegetation production decreases in the tropics whereas it increases in temperate regions. We identify regions in which these El Niño-induced changes are consistent with potential state transitions in global terrestrial ecosystems, including potential greening of western North America, dieback of the Amazon rainforest, and further aridification of south-eastern Africa and Australia.

## 1 Introduction

Earth's current climatology can be considered a steady state of a non-equilibrium system (Kleidon, 2010) in which constant influx of solar radiation and internal feedback mechanisms stall equilibrium, so that properties of the climate system can remain relatively close to mean values. However, it is becoming increasingly clear that the Earth's climate system as a whole, as well as some of its large-scale components or subsystems, have tipping points (i.e. critical thresholds in forcing and/or a feature of the system) at which the system shifts from one state (e.g. current climate) to another (e.g. future climate) (Lenton et al., 2008; Scheffer et al., 2009). Perturbations related to human-induced climate change could push several Earth's subsystems past a tipping point (Lenton, 2011; Barnosky et al., 2012), potentially changing the modes of natural climatic variability (Cai et al., 2015).

Lenton et al. (2008) identified several *tipping elements* in the Earth's climate system. A tipping element is a large-scale component of the Earth system that may pass over a tipping point with the potential to alter global climate. A pronounced change in the amplitude and/or frequency of the El Niño–Southern Oscillation (ENSO), including its shifting into a permanent

El Niño state, is among the policy-relevant tipping elements identified by Lenton et al. (2008). Although there is no definitive evidence at present for specific changes in ENSO behaviour in response to climate change (McPhaden et al., 2006; Cai et al., 2015), global climate modelling studies do indicate the possibility of ENSO intensification during the 21st century due to anthropogenic forcing (Timmermann et al., 1999; Guilyardi, 2006). Greenhouse warming in climate projections can lead to more frequent El Niño events, because it diminishes temperature gradients in the eastern equatorial Pacific Ocean, which facilitate the changes in convection zones that are observed during this phenomenom (Cai et al., 2014; Latif et al., 2015).

Through atmospheric teleconnections, ENSO exerts strong effects on manifold environmental and social systems at global to regional scales (Diaz and Markgraf, 2000; Cai et al., 2015; Yeh et al., 2018; McPhaden et al., 2006), including major implications for the functioning of terrestrial ecosystems (Holmgren et al., 2001). For instance in Amazonia, temperature and precipitation changes associated with ENSO have been found to alter the net carbon flux to the atmosphere and seasonal inundation of floodplains (Foley et al., 2002). Moreover, El Niño-induced droughts in Amazonia increase the likelihood of forest fires that modify landscape features and have top-down effects that damage not only vegetation but also faunal populations (Alencar et al., 2006). Similarly, in Australia's drylands, ENSO is a modulator of biomass removal through fires and grazing (Holmgren et al., 2006). Such land-use impacts are particularly important in places where the population largely depends on agriculture, such as south-eastern Africa, where ENSO also affects rainfall patterns (Indeje et al., 2000). Therefore, and despite the inherent uncertainty of climate change projections, the potential impact of shifting the state of the ENSO is of global concern (Lenton et al., 2008; Cai et al., 2014). Advancing the quantitative understanding of the range of expected physical climate changes (let alone tipping events) and their potential impact on different sectors and/or regions is crucial for decision-making related to global change (Lenton and Ciscar, 2013; Zebiak et al., 2015).

Here we used the Planet Simulator (PlaSim; Fraedrich et al., 2005a, b) to investigate the potential impact on global climate of shifting the dynamics of the ENSO into a permanent El Niño state, and examine the potential consequences of this tipping event on global terrestrial ecosystems. Our aim was to see what Earth's climate could be if the climatology of sea surface temperature (SST) were that observed during a warm phase of ENSO. Thus we simulated a control scenario and compared it to current climate using data sets derived from observations, in order to understand the model's limitations. Then we set up a permanent El Niño experiment and compared it to said control simulation, focusing on the impacts on water deficit and terrestrial ecosystems. We also briefly discuss how this permanent El Niño experiment relates to data and simulations from the warm Pliocene period, when it has been suggested that sustained El Niño conditions occurred (Wara et al., 2005). The exploratory nature of our modelling approach is more about identifying potential impacts than about estimating their likelihood.

## 2 Model description

PlaSim is a climate model of intermediate complexity. This kind of models is well-suited for our goals because of their ability to parsimoniously simulate feedback mechanisms between many components of the climate system over long-term simulation ensembles covering a global domain (Claussen et al., 2002). The dynamical core of PlaSim is based on PUMA (Fraedrich

et al., 2005c), an atmospheric general circulation model that solves the moist primitive equations representing the conservation of momentum, mass and energy in the Earth's atmosphere. It uses a vertical $\sigma$ coordinate and the spectral transform method with triangular truncation over a sphere (Orszag, 1970; Hoskins and Simmons, 1975). Unresolved sub-grid scale processes such as radiation, moist and dry convection, large-scale precipitation, surface fluxes and vertical and horizontal diffusion are parameterized (Lunkeit et al., 2011).

The atmospheric model is coupled to a five layer land surface model that includes parameterizations for soil temperature, snow cover and a bucket-type runoff transport scheme (Sausen et al., 1994). A dynamic vegetation model (SimBA) included in PlaSim allows to represent vegetation-atmosphere feedback mechanisms through climate-induced changes in large-scale surface parameters such as albedo, roughness and water holding capacity (Kleidon, 2006). The ocean can be represented through prescribed climatological values of SST or using dynamical models (Fraedrich, 2012; Semtner Jr, 1976). For more details about the model description we refer the reader to Fraedrich et al. (2005a) and Lunkeit et al. (2011).

PlaSim has been successfully used to study the evolution of the Earth's climate system under different physical scenarios and forcings. One of the first applications of PlaSim was to estimate the maximum possible influence of vegetation on the global climate through simulation of two extreme scenarios: "green planet" and "desert world" (Fraedrich et al., 2005b). More recent applications have studied the planetary climate system's response to, for instance, vegetation-atmosphere feedback mechanisms (Dekker et al., 2010; Bathiany et al., 2012), large-scale changes in orography (Garreaud et al., 2010; Schmittner et al., 2011), and variations in astronomical forcings (Boschi et al., 2013; Linsenmeier et al., 2015). The model has also been used for paleoclimate studies (Henrot et al., 2009) and snowball Earth experiments (Spiegl et al., 2015; Lucarini et al., 2013). Some of these studies have also explored the existence of alternative equilibrium states of the planetary climate system, as well as the mechanisms behind potential state transitions (Dekker et al., 2010; Bathiany et al., 2012; Boschi et al., 2013; Linsenmeier et al., 2015).

## 3 Experimental setup

For all simulations, PlaSim was implemented with a spectral horizontal resolution of T21 (about $5.6°$ on a Gaussian grid) and 10 non-equally spaced $\sigma$ vertical levels. Though relatively coarse, this resolution allows a realistic representation of large-scale atmospheric circulation (Linsenmeier et al., 2015) and baroclinic structures that are important in the energy transport poleward in a slowly rotating and phase locked planet like Earth (Lucarini et al., 2013). All simulations were integrated for 500 years at 40 minutes time steps, with model output being stored as daily averages. Such long-term simulations allow the model to approach an equilibrium state (Lucarini et al., 2017), which we characterized using the last 30 years of each simulation. The dynamic vegetation model SimBA was used to allow variations in terrestrial ecosystems. The solar constant value was set to $1365\,\mathrm{W\,m^{-2}}$. For atmospheric $CO_2$ we used values $360\,\mathrm{ppm}$ and $415\,\mathrm{ppm}$, however, differences among simulations were modest enough that we present here only the results obtained with the latter, which is closer to current estimates of this variable (Dlugokencky and Tans, 2019). All other model physical constants, planetary values and tuning parameters were kept at their default setting (Fraedrich et al., 2005b).

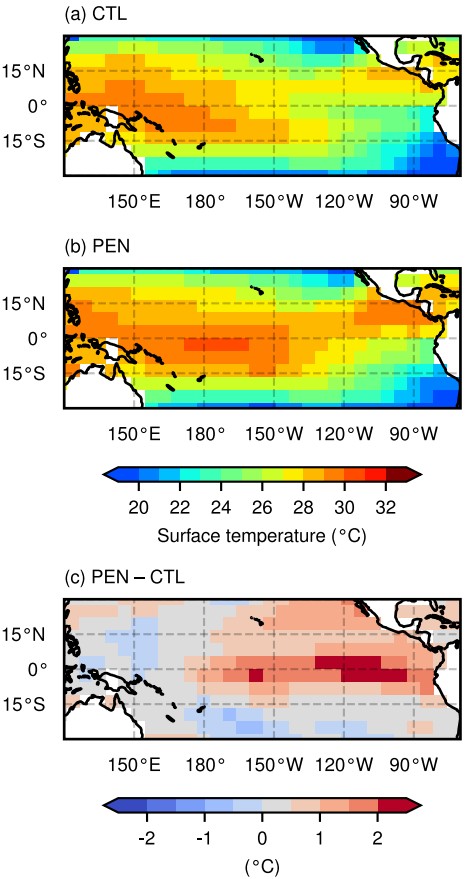

**Figure 1.** Annual mean of prescribed SST climatology in the equatorial Pacific Ocean region for CTL (a) and PEN (b) simulations, and their absolute differences (c).

The ocean was represented through prescribed SST values that were used to define two scenarios: ConTroL (CTL) and Permanent El Niño (PEN). Differences between scenarios are entirely determined by differences in the prescribed SST annual cycle (Figs. 1 and S1). Due to the inherent chaotic nature of atmospheric dynamics, each scenario was studied using an equally weighted ensemble of three simulations with different initial conditions, which were varied through a random noise disturbance in the surface pressure field (e.g. Kunz et al., 2009). Though we found little sensitivity to these variations in initial conditions (Fig. S2), we nevertheless used the ensemble means in our analyses. Average results from different scenarios were compared using a Student's $t$-test at a significance level of $\alpha = 0.05$ and 30 degrees of freedom (corresponding to a 30 year simulation period).

## 3.1 Control scenario and model evaluation

The CTL scenario is based on present-day climatology and therefore was first used to assess the model's ability to realistically represent the Earth's climate system, and then used as reference for comparison with the PEN scenario. The CTL scenario is characterized by a climatological annual cycle of SST values obtained from the second phase of the Atmospheric Model Inter-comparison Project (AMIP II) for the period 1979–2010 (Hurrell et al., 2008). Regridding to model resolution and adjusting the least possible temperature to be $271.38\,\mathrm{K}$ was done following recommendations by Taylor et al. (2000).

Observational gridded data sets used to compare with the model results for the CTL scenario are: 1987–2016 monthly values of precipitation from the Global Precipitation Climatology Project (GPCP; Adler et al., 2003); 1988–2017 monthly values of near surface air temperature from the HadCRUT4 data set (Morice et al., 2012); 1979–2004 monthly values of mean sea level pressure from the HadSLP2 data set (Allan and Ansell, 2006); 2000–2015 monthly values of gross primary productivity (GPP) from the MODIS product MOD17A2 (Zhao et al., 2005); and 1987–2016 monthly values of free air temperature, as well as zonal and meridional winds at different pressure levels ($1000$–$50\,\mathrm{hPa}$) from the European Centre for Medium-range Weather Forecasts (ECMWF) Interim reanalysis (ERA-Interim; Dee et al., 2011). The global energy and water balances produced by PlaSim were compared with estimates based on ERA-Interim and those reported by Trenberth et al. (2011). More details about all data sets used in this work are shown in Tables S1 and S2 as part of the supplementary material.

## 3.2 Permanent El Niño

In this scenario the prescribed climatological annual cycle of SST was replaced with the annual cycle of El Niño occurred between June 2015 and May 2016, using monthly values from the AMIP II data set. This is the strongest El Niño recorded so far, surpassing strong events from last century: 1982–1983 and 1997–1998 (Jiménez-Muñoz et al., 2016). The bottom panel (or c) in Fig. 1 shows the annual mean differences between the PEN and CTL scenarios for the equatorial Pacific region. It can be seen that this El Niño event depicted well the first main mode of spatio-temporal variability of SST anomalies (the mature phase of ENSO) and did not exhibit the El Niño Modoki second main mode (Ashok et al., 2007). The PEN climatology has a global annual warm bias of $+0.4\,^{\circ}\mathrm{C}$, with both boreal winter (DJF) and autumn (SON) seasons having the largest positive deviations (Fig. S1). The warm bias is greater specifically in the equatorial Pacific Ocean, where it reaches an annual mean value of $+0.8\,^{\circ}\mathrm{C}$.

Wara et al. (2005) suggest the possibility that permanent El Niño-like conditions have occurred during the warm early Pliocene period. Using environmental reconstructions from isotopes and bioindicators, they found that the zonal west-to-east gradient of SST stayed very close to the one observed during modern El Niño events. Although more recent studies suggest that their reconstruction may underestimate the Pacific warm pool temperature and its variability (Zhang et al., 2014), it is still interesting to compare PEN scenario with data from the warm Pliocene period. Thus, we compared the SST forcing of PEN with the paleoenvironmental reconstruction data set PRISM3 (Dowsett et al., 2009), which covers said period and has been used in several Pliocene modelling experiments (Haywood et al., 2016). PEN forcing is on average about $1\,^{\circ}\mathrm{C}$ warmer in the tropics, though differences in this region are not all significant (Fig. S3). PEN is cooler than PRISM3 elsewhere. The

west-to-east zonal gradient was computed for CTL and PEN scenarios, as well as for PRISM3, at the sites shown in Fig. S3. For PEN the gradient is $1.5\,^{\circ}$C, whereas for PRISM3 it is $1.8\,^{\circ}$C, and for CTL it is $3.1\,^{\circ}$C. Since this gradient is a good indicator of strength of the Walker circulation (Wara et al., 2005), it is expected that circulation will be weaker in PEN than in the mid-Pliocene and much weaker than in the CTL experiment.

## 3.3 Climatological water deficit estimation

In this study we focus on the consequences that such a PEN climate scenario could have for terrestrial ecosystems. We used the maximum climatological water deficit (MCWD) as defined by Aragao et al. (2007), which is an indicator of accumulated water stress during dry seasons and has been linked to ecosystem degradation via hydraulic failure in plants (Leitold et al., 2018), fires (Brando et al., 2014) and deforestation (Costa and Pires, 2010). We followed an approach similar to Malhi et al. (2009), who plot MCWD versus mean annual precipitation (MAP) to study the possibility of land cover transitions under climate change projections in the Amazon rainforest. Malhi et al. (2009) found that most savannah biomes in Amazonia (i.e. Cerrado), had rainfall rates below $1500\,\mathrm{mm\,year^{-1}}$ and accumulated water deficits less than $-300\,\mathrm{mm}$. Unlike Malhi et al. (2009), our aim was global and, therefore, in order to be able to calculate MCWD, we could not use a single fixed evapotranspiration rate (ET) value for the full extent of our domain. Instead we used annual mean ET for every gridpoint from the CTL scenario as a proxy of expected ET with current land cover. These reference values (Fig. S4) agree with the approximation used by Malhi et al. (2009) of $100\,\mathrm{mm\,month^{-1}}$ average ET for Amazonia. Likewise, we assumed soil saturation at the wettest time of the year for each gridpoint and set the cumulative water deficit to zero at this month (Fig. S5).

Additionally, we judged useful to combine, in a single global map, statistically significant differences between scenarios in MAP and MCWD, in order to support our discussion about the impact that a PEN scenario could have on terrestrial ecosystems. For this map, we defined $10\,\%$ of the CTL value as a threshold to identify places where differences between scenarios were the largest. In this way, what we mean by "much greater/less", is that the difference exceeds said threshold. Otherwise, even though both MAP and MCWD changed *significantly* for a particular gridpoint, this change was not as pronounced and only "greater/less" are used. Different colours represent the possible combinations of changes in both variables.

## 4 Results

### 4.1 Control climate

A comparison between PlaSim results for the CTL scenario and estimates from ERA-Interim and Trenberth, Fasullo, and Mackaro (2011, henceforth TFM11) shows that simulations produce realistic estimates of the global energy and water budgets (Fig. 2). At the top of the atmosphere (TOA), the magnitude of the net energy imbalance (i.e. the difference $\Delta$ between incoming and outgoing radiation) is lower for PlaSim ($\Delta = +0.4\,\mathrm{W\,m^{-2}}$) than it is for ERA-Interim ($\Delta = -1.4\,\mathrm{W\,m^{-2}}$) and TFM11 ($\Delta = +0.9\,\mathrm{W\,m^{-2}}$). Previous studies have reported that PlaSim's energy imbalance is closer to zero than it is for many state-of-the-art climate models (Lucarini et al., 2010). Differences between PlaSim and observations indicate that

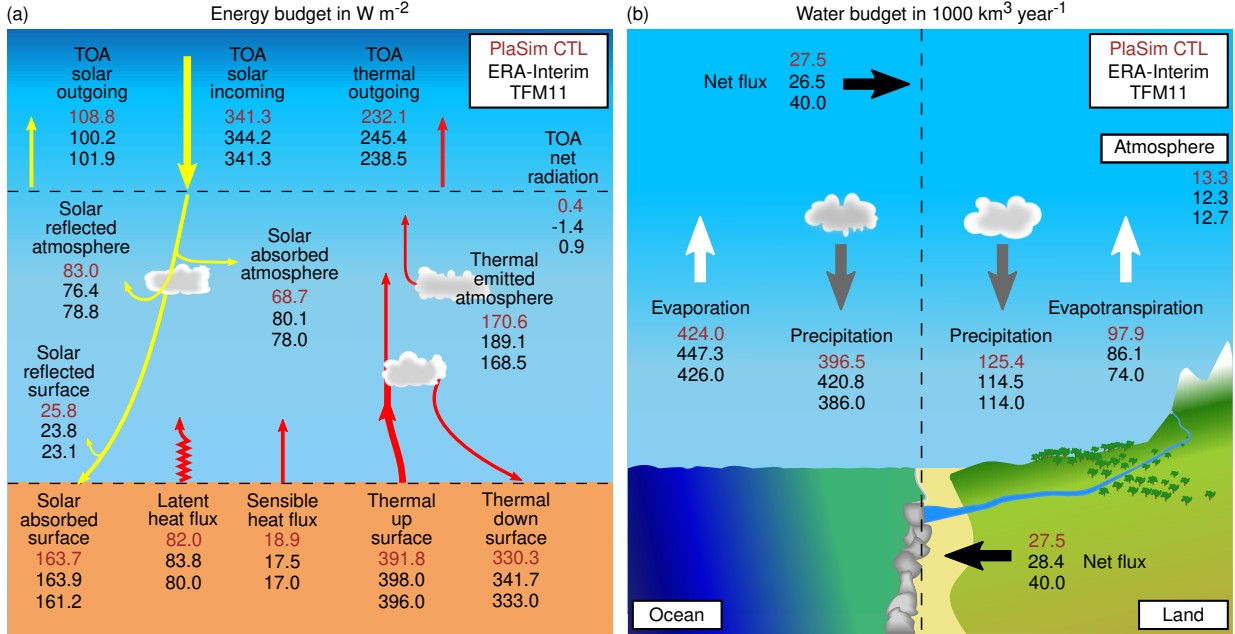

**Figure 2.** Global average annual mean energy (a) and water (b) budgets for the CTL simulation compared with ERA-Interim and Trenberth, Fasullo, and Mackaro (2011). Units are shown in the title of each panel.

the model overestimates the outgoing solar radiation at TOA, as a result of more solar radiation being reflected both at the surface and atmosphere, and less solar radiation being absorbed by the atmosphere. However, the model's estimate of solar radiation absorbed at the surface falls in between those of ERA-Interim and TFM11. The surface energy balance in the model shows realistic estimates of thermal, latent heat (LH) and sensible heat (SH) fluxes, with a slightly overestimated Bowen ratio

(SH/LH) of 0.23 as compared to 0.21 in both ERA-Interim and TFM11. These biases in the surface fluxes are likely to be related to prescribed values of aerodynamic roughness length (Yang et al., 2002), which has also been related to ERA-Interim biases (Zhou and Wang, 2016). The model's representation of the greenhouse effect through the estimate of thermal energy emitted by the atmosphere falls also within estimates from ERA-Interim and TFM11.

As for the water budget, PlaSim presents a sensible distribution of water among ocean, land and atmosphere, preserving

key hydrological cycle characteristics: greater evaporation than precipitation over the oceans and greater precipitation than evapotranspiration over continental areas (Fig. 2b). As compared to TFM11, atmospheric transport from ocean to land, as well as continental runoff, are underestimated both in PlaSim and ERA-Interim, but unlike the reanalysis, the water balance for the model shows conservation. This is partly because the model values are not affected by any assimilation scheme, as is the reanalysis (Seager and Henderson, 2013). Overall, PlaSim estimates of global water fluxes fall within the uncertainty ranges

proposed by TFM11 for ERA-Interim and other reanalyses.

There are statistically significant differences between PlaSim and observational data in the global annual mean distribution of near surface air temperature (Fig. 3c), precipitation rate (Fig. 3f), GPP (Fig. 3i) and mean sea level pressure (Fig. 3l). However,

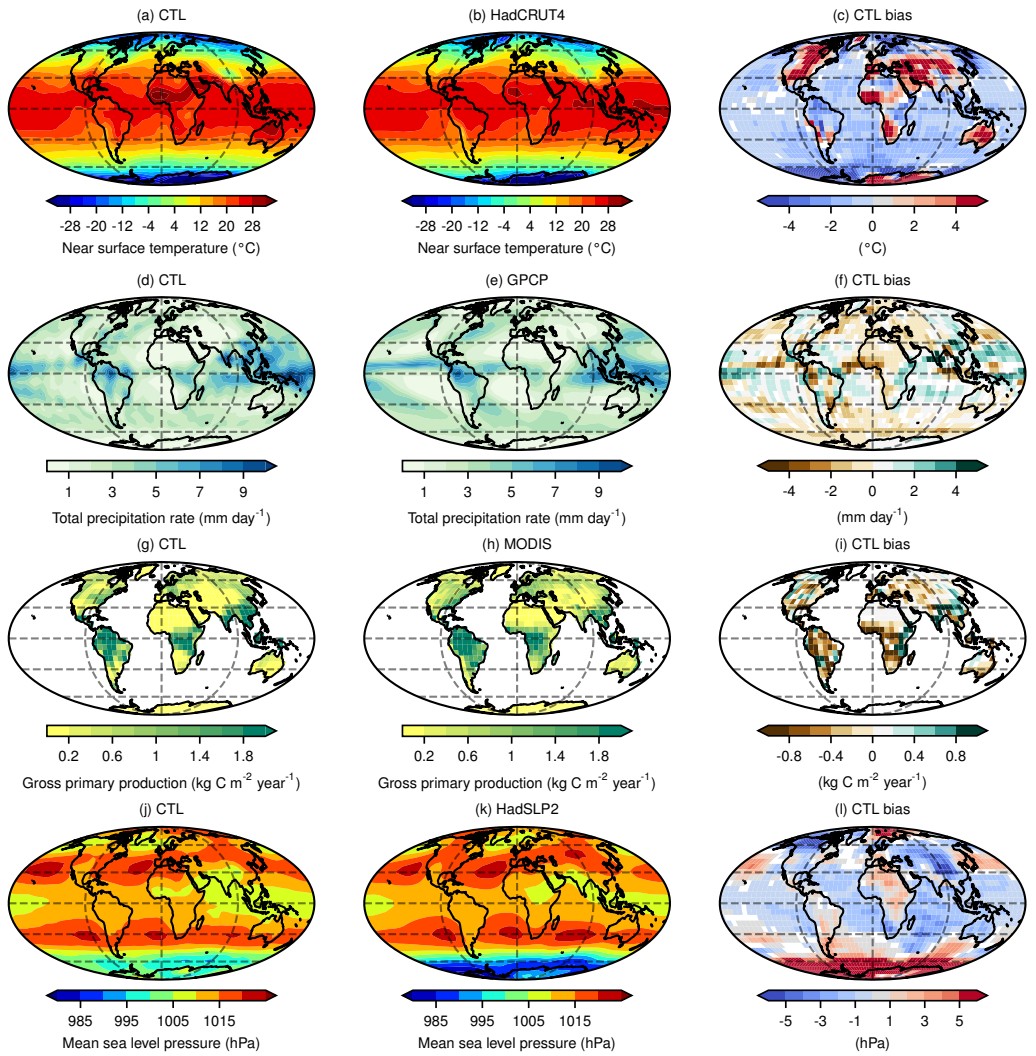

**Figure 3.** Annual mean results for the CTL simulation compared with observational data: near surface temperature in CTL (a) and Had-CRUT4 (b) and model bias (c); total precipitation in CTL (d) and GPCP (e) and model bias (f); GPP in CTL (g) and MODIS (h) and model bias (i); and mean sea level pressure in CTL (j) and HadSLP2 (k) and model bias (l). In the bias panels, white depicts gridpoints with statistically non-significant difference ($\alpha = 0.05$). Grid lines are spaced every $30°$ from the Equator and $90°$ from Greenwich, in latitude and longitude respectively.

these biases are within typical ranges obtained for more comprehensive state-of-the-art climate models (Stevens et al., 2013), and cannot be entirely attributed to model deficiencies because of uncertainties in the observational data sets themselves (e.g. Prein and Gobiet, 2017). Importantly, large-scale spatial patterns of near surface air temperature, precipitation rate, GPP and mean sea level pressure (Fig. 3), as well as their seasonality (Figs. S6–9), are realistically represented by PlaSim.

Differences between PlaSim results and HadCRUT4 (Fig. 3a, b, c) indicate that the model exhibits a global average cold bias of about $-0.4\,°C$. This global bias combines an average warm bias of $+0.9\,°C$ over continental areas and an average cold bias of $-1.4\,°C$ over the oceans. The global average annual mean temperature in the CTL simulation ($13.9\,°C$) is similar to that from observations ($14.3\,°C$). Seasonal variations of temperature are sensibly simulated too (Fig. S6). Some of the largest temperature biases occur on land near polar regions, especially in Greenland and Antarctica, and regions with complex terrain such as the Himalayas mountain range. Biases over complex terrain are related to known limitations even in state-of-the-art climate models, in part because the orography representation has to be smoothed to keep dynamical stability (Flato et al., 2013). Warm biases in western Africa and Australia and slightly cold biases in the Sahara desert and Amazonia are also observed. Strongest biases take place in boreal summer (JJA) and autumn (SON) seasons, when the global average biases on land are $+1.7\,°C$ and $+1.6\,°C$ respectively. Over the oceans, temperatures from the model are colder than observations year-round, with a maximum of $-1.6\,°C$ in boreal winter (DJF).

Annual mean daily precipitation from the CTL simulation is depicted in Fig. 3d. The large-scale patterns exhibited in GPCP (Fig. 3e) are present: the equatorial maximum in the Inter-Tropical Convergence Zone (ITCZ), middle latitudes peaks of synoptic-scale weather systems and low precipitation rates in the polar regions. There are high precipitation rates over the western equatorial Pacific and the Western Hemisphere warm pools. The PlaSim global average annual mean precipitation rate reaches $2.8\,\mathrm{mm\,day^{-1}}$, whereas for observational data it is $2.7\,\mathrm{mm\,day^{-1}}$. Concerning seasonal variation (Fig. S7), tropical precipitation shows latitudinal displacement and has maximum peaks in south-eastern Asia in the boreal winter (DJF) and summer (JJA) seasons. In northern South America the maximum of precipitation crosses the equator twice a year, unlike over the eastern Pacific where the ITCZ stays north of the equator for all seasons. PlaSim also displays the double ITCZ band that has been described for other global climate models (Lin, 2007), especially during the boreal spring season (MAM).

Figure 3g shows the dynamic vegetation results of CTL for GPP on continental areas. Global average annual mean for the CTL simulation reaches $0.7\,\mathrm{kg\,C\,m^{-2}\,year^{-1}}$, whereas for MODIS data it is $0.8\,\mathrm{kg\,C\,m^{-2}\,year^{-1}}$. Although large deviations with respect to MODIS can be seen in Fig. 3i, these are similar to those reported in other modelling studies and could be related to the climate data sets used as input in the simulations (Ardö, 2015; Wu et al., 2018). Differences between this model and other data sets (whether derived from observations or not) occur because parameterization schemes to obtain GPP indirectly from climate variables are different. Nevertheless, the model shows a realistic estimate of the GPP distribution with high productivity near the equator and low productivity elsewhere. Also the location of the subtropical deserts is consistent with observations (Fig. 3h). Furthermore, the model is able to represent seasonal changes in primary productivity (Fig. S8). There is a year-round high productivity around the equator, whereas for the Northern and Southern Hemispheres it is maximum in their respective summer season and it is minimum in their respective winter season.

When compared with observations, PlaSim also displays a very similar annual mean pattern of high and low pressure cells in the subtropics (Fig. 3j, k). The largest differences are seen near Antarctica, where the model overestimates sea level pressure (Fig. 3l). Seasonally, the model is also able to represent the appearance of semi-permanent pressure systems throughout the year (Fig. S9). For instance, in the Northern Hemisphere, the Aleutian and Icelandic lows and the Pacific, Azores and Siberian highs are seen in winter, whereas only the Pacific and Bermuda highs are seen in the summer season.

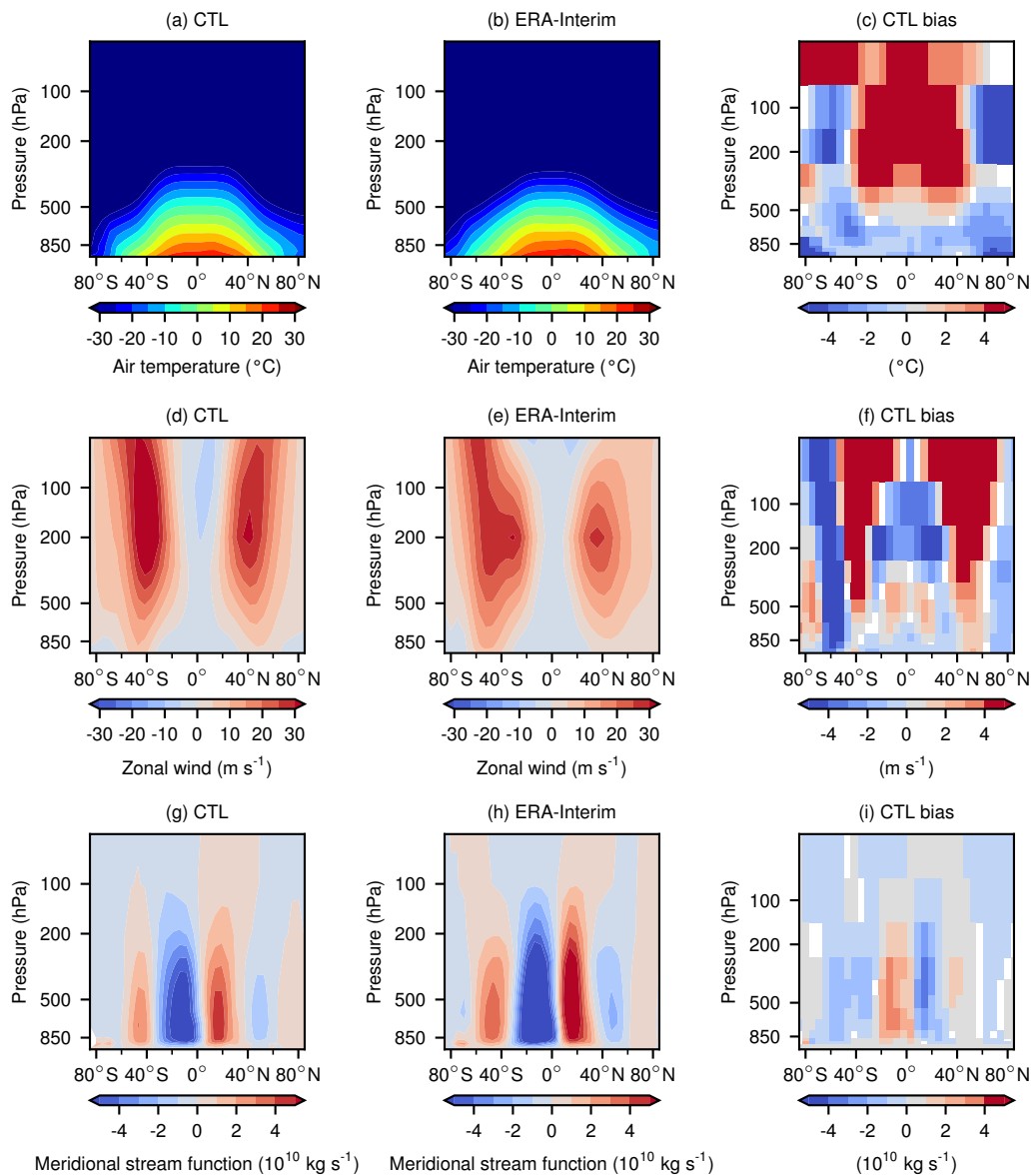

**Figure 4.** Zonal average annual means results for the CTL simulation compared with ERA-Interim at different pressure levels for: air temperature in CTL (a), ERA-Interim (b) and model bias (c); zonal wind in CTL (d), ERA-Interim (e) and model bias (f); and meridional mass stream function in CTL (g), ERA-interim (h) and model bias (i). In the bias panels, white depicts gridpoints with statistically non-significant difference ($\alpha = 0.05$).

Regarding general circulation, the temperature gradient in the atmosphere for the CTL simulation has a very similar structure as the one from ERA-Interim reanalysis (Fig. 4a, b). Differences are relatively small below $500\,\mathrm{hPa}$ and away from the poles

(Fig. 4c). Strongest biases are found around $200\,\mathrm{hPa}$. In the upper troposphere, the model is much warmer in the tropics and much cooler near the North Pole. This behaviour is common to most general circulation models, which have difficulties parameterizing moist thermodynamic processes (Santer et al., 2017). Convective heating is overestimated in the Kuo-type convection scheme implemented in PlaSim (Arakawa, 2004).

The distribution of temperature is closely related to the zonal wind velocity (Fig. 4d, e). The location of the sub-tropical jet streams in the CTL simulation is in agreement with ERA-Interim, with the zonal wind speeds of the jet streams being higher in PlaSim. As with temperature, main differences are found in the upper troposphere and the lower stratosphere above $200\,\mathrm{hPa}$ (Fig. 4f). This is a known problem with some general circulation models that display too fast zonal jet streams due to a coarse representation of the planet's orography and simple representation of the low-level drag (Pithan et al., 2016). The trade winds
in PlaSim prevail around the equator, but are slower than in the reanalysis below $300\,\mathrm{hPa}$, and faster above. As compared to ERA-Interim, zonal wind speed is underestimated eastward in PlaSim at $60°$ S at all pressure levels.

PlaSim adequately simulates the thermally induced Hadley cell and its displacement across the equator for the annual mean (Fig. 4g, h), with clockwise rotation when in the Southern Hemisphere and counter-clockwise in the Northern. As described for observations, when in the Southern Hemisphere, the cell is stronger. Similarly, the indirect Ferrell cell can be seen moving
in the opposite direction as the former (different colour in Fig. 4g, h). Differences in the stream function occur for the Hadley cell at both hemispheres, which is slower and shorter in PlaSim (Fig. 4i). Because of this, the indirect Ferrell cell is also smaller and is almost absent when in the Northern Hemisphere. Polar cells are present but not visible in any of these figures because of the chosen contour levels.

Altogether, the PlaSim CTL simulation realistically represents the broadscale patterns seen in observational data of present-
day climate. In spite of PlaSim's simplicity and coarse resolution of simulations, the model produces a sensible picture of the current climatological state of the Earth's climate system. Our results generally agree with previous studies showing that PlaSim is able to simulate many dominant features of the present-day climate (e.g. Garreaud et al., 2010; Lucarini et al., 2017). Global averages of annual mean and seasonal mean values for CTL and observational data sets are also summarized in Table S3.

## 4.2   Permanent El Niño climate

To be able to judge this scenario on account of its different ENSO state with respect to CTL, we checked that model bias was similar in PlaSim during El Niño conditions and CTL. Consequently, we compared the PEN scenario to the observational data sets during very strong El Niño events, except for MODIS data which only includes moderate ones (Fig. S10). Model bias for PEN variables is very similar to that shown in Fig. 3, and also there is good agreement in the broadscale patterns seen in PEN
and observations during El Niño events. Figure S11 shows more clearly that model bias is very similar in both experiments, hence we can attribute most of the differences between CTL and PEN to their ENSO condition.

Warmer than usual equatorial Pacific Ocean temperatures, as set in the PEN scenario, lead to statistically significant changes in the global energy and water budgets (Fig. 5). For the energy balance, a warmer ocean surface results in higher long-wave radiation being emitted, as seen in the heat fluxes and the thermal upward radiation. It also diminishes surface snow and sea

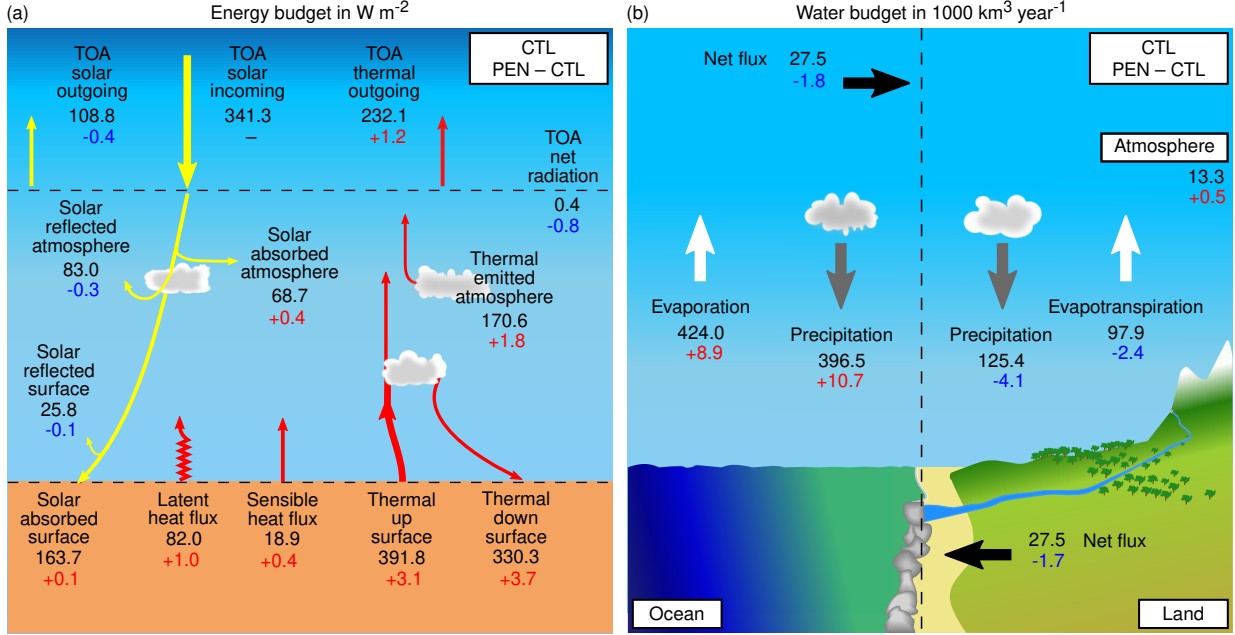

**Figure 5.** Global average annual mean energy (a) and water (b) budgets for the PEN simulation compared with the CTL simulation. Units are shown in the title of each panel. En dashes (–) show statistically non-significant differences ($\alpha = 0.05$).

ice formation, which is followed by a low planetary effective albedo and, therefore, less solar incoming radiation is reflected. Like the ocean, the atmosphere also becomes warmer and its thermal emission increases. Ultimately this leads to less incoming solar radiation being reflected, and more long-wave radiation being emitted back to space. These two processes do not balance each other in the model and, therefore, the net imbalance at TOA becomes slightly more negative in PEN than in the CTL
simulation (Fig. 5a). As expected, the PEN scenario shows much warmer surface and atmosphere than CTL.

According to PEN simulation outputs, water is redistributed throughout the world (Fig. 5b). A warmer atmosphere stores more water, but the net transport of water from the ocean to land is reduced. This may be explained by changes in wind patterns over the Pacific Ocean, linked to the Walker circulation, which keep most water vapour above the ocean surface, as is typical during El Niño conditions (Diaz and Markgraf, 2000). Consistently, evaporation and precipitation increase over the ocean while
they decrease on land. The values in Fig. 5 are also presented in tabular form in Tables S4 and S5, including the results of the significance test.

There are statistically significant differences in the near surface air temperatures for the PEN scenario as compared to the CTL simulation (Fig. 6a). PEN has an average global annual mean increase of approximately $0.5\,°C$. This temperature change is greater on land with an average $0.8\,°C$ increase, than over the oceans, where the change reaches $0.5\,°C$. In the tropics,
variation occurs mainly in the equatorial Pacific region (as expected), but also in Central America, northern South America, Amazonia, the Sahel region in Africa, and northern Australia. Moreover, air over the North Atlantic Ocean is slightly colder, while the Arctic Ocean near Canada and Siberia is warmer. Air over some ocean areas in the Southern Hemisphere, close to

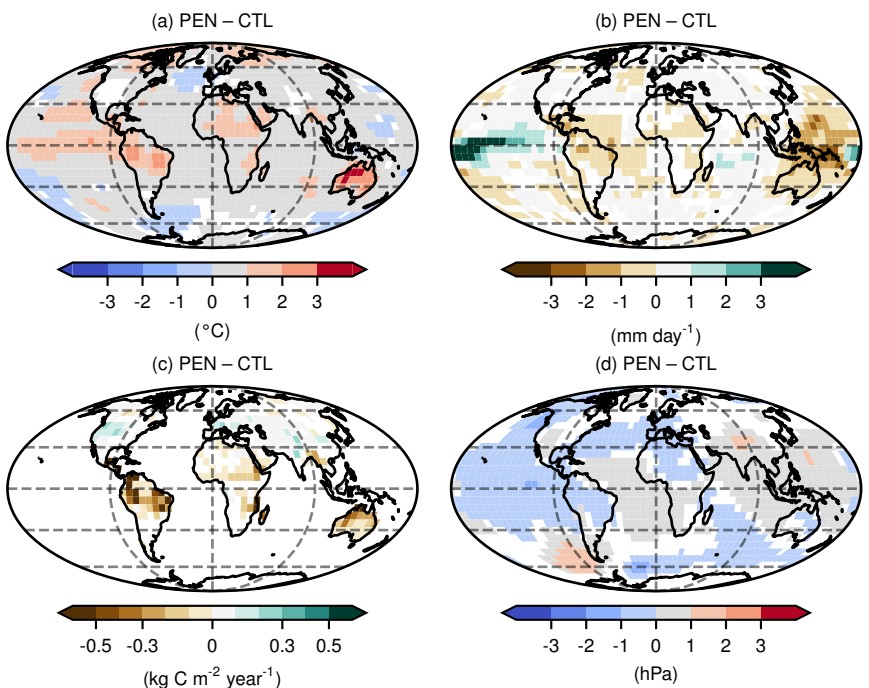

**Figure 6.** Differences in annual means results for the PEN simulation compared with CTL simulation (PEN minus CTL) for: near surface temperature (a), daily precipitation (b), GPP (c) and mean sea level pressure (d). In all panels, white depicts gridpoints with statistically non-significant difference ($\alpha = 0.05$). Grid lines are spaced every $30°$ from the Equator and $90°$ from Greenwich, in latitude and longitude respectively.

Antarctica, becomes colder in PEN than in CTL. No significant differences in temperature were seen in most of North America. Seasonally, boreal winter (DJF) shows the greatest warming on land, especially for Australia (Fig. S12). South America is the warmest during boreal summer (JJA).

Regarding precipitation, PEN leads to an average rise in global mean precipitation of approximately $0.1\,\mathrm{mm\,day^{-1}}$ and a
5   decrease in continental areas of about the same value (Fig. 6b). Precipitation over the tropical Atlantic Ocean is also slightly reduced in the PEN scenario. On land the most conspicuous differences occur near the equator, in Central America and in Amazonia, where precipitation is reduced. Conversely, in the Northern Hemisphere above midlatitudes, land areas experience slightly higher precipitation rates. Similar to temperature, boreal winter (DJF) exhibits the largest deviations (Fig. S12), with above (below) CTL rainfall in equatorial eastern (western) Pacific Ocean.

10    Higher temperatures and precipitation rates in temperate zones result in higher productivity rates (Fig. 6c). The opposite happens in regions where temperatures increase while precipitation rates decrease, as in Central America and northern South America and Amazonia. In Africa, near the Middle East, there is a relatively small increase in productivity related to increased precipitation. On the contrary, below the Sahel region and south eastern Africa, GPP diminishes where precipitation

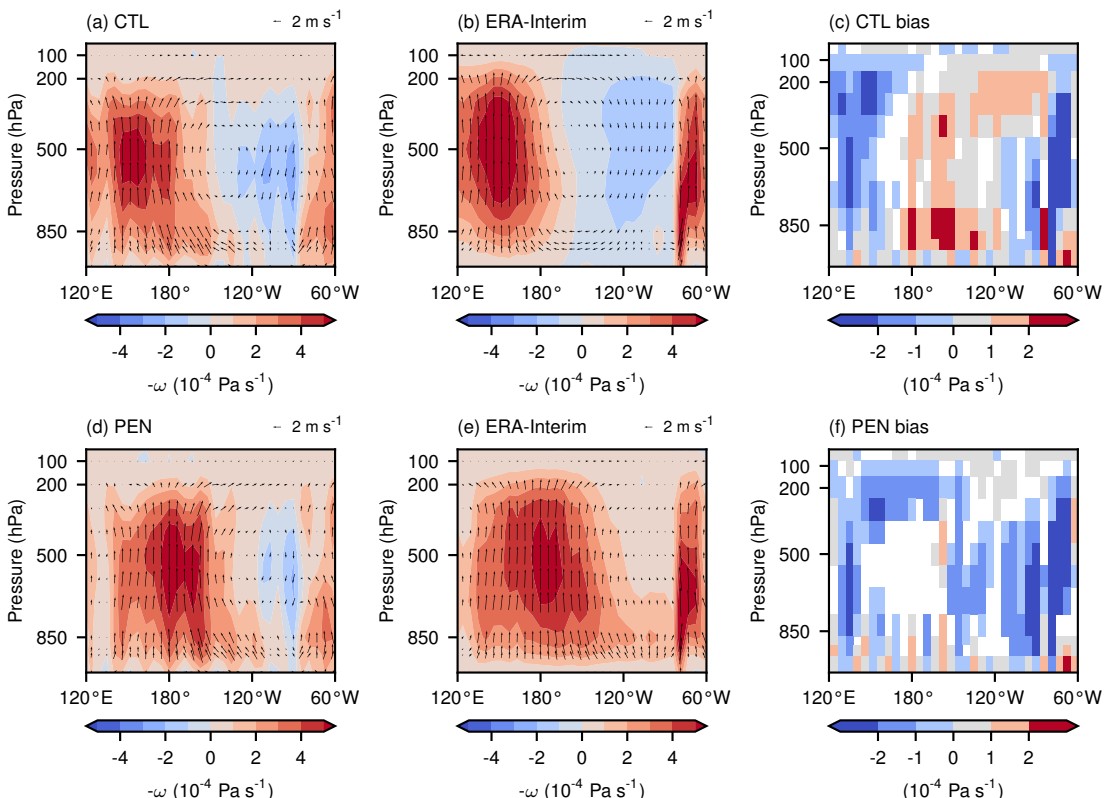

**Figure 7.** Meridional divergent circulation in the equatorial Pacific for CTL simulation (a), climatological ERA-Interim (b) and CTL bias in vertical velocity values (c); as well as for PEN simulation (d), ERA-Interim mean of 1997–1998 and 2015–2016 (e), and PEN bias in vertical velocity values (f). Vectors are plotted using the divergent meridional wind component and the negative vertical velocity in pressure units ($-\omega$), averaged over latitudes 5° S to 5° N. Filled contours show the magnitude of the vertical velocity. In all bias panels white is for statistically non-significant differences ($\alpha = 0.05$).

declined with respect to the CTL simulation. The same happens in Australia. Globally, a decrease in productivity of about $0.1\,\mathrm{kg\,C\,m^{-2}\,year^{-1}}$ occurs in the PEN scenario. PEN results also show decreased productivity in South America year-round (Fig. S13).

PEN simulation shows the Southern Oscillation hemispheric zonal dipole in surface pressure (Fig. 6d). In the tropics, there
5  is an overall increase in surface pressure over the oceans (except in the eastern Pacific), which is related to drier air and reduced transport of moisture into continental areas. As expected, lower (higher) surface pressure conditions remain year-round in eastern (western) equatorial Pacific Ocean (Fig. S13). The North Atlantic region has lower surface pressure throughout the year, but most markedly in boreal winter (DJF).

The atmosphere becomes warmer in PEN than it is in the CTL simulation at most pressure levels and latitudes (Fig. S14a).
10  There is only a slight cold bias in the Northern Hemisphere above $200\,\mathrm{hPa}$. The hot spot in the tropics in the upper troposphere

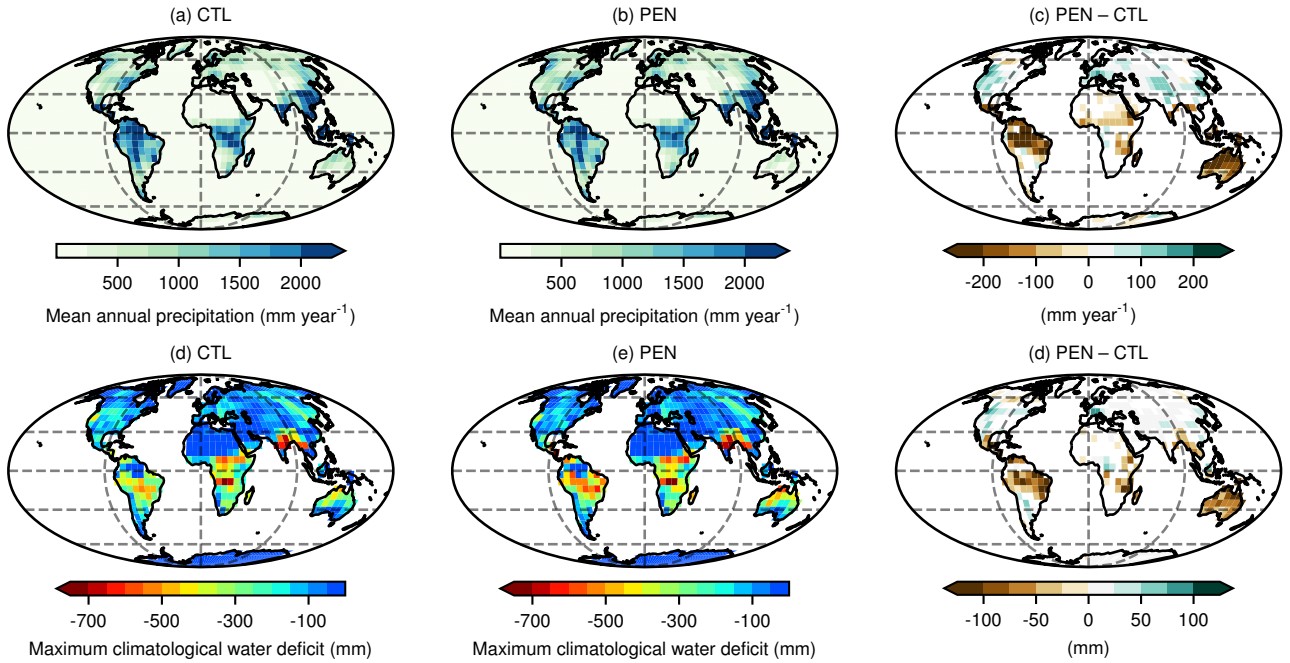

**Figure 8.** Annual mean results for the PEN simulation compared with the CTL simulation for: MAP in CTL (a) and PEN (b) and their absolute difference (c), and MCWD in CTL (d) and PEN (e) and their absolute difference (f). In the difference panels, white depicts gridpoints with statistically non-significant difference ($\alpha = 0.05$). Grid lines are spaced every $30°$ from the Equator and $90°$ from Greenwich, in latitude and longitude respectively.

is much warmer in the PEN simulation. This result has been previously reported and seems to be related to changes in the radiative forcing of the troposphere layer (Lin et al., 2017). Wind speed increases eastward (or lessens westward) in the model atmosphere (Fig. S14b). Notably, in the midlatitudes in the middle troposphere, the zonal velocity rises eastward, moving the sub-tropical jet streams towards the equator as commonly seen during El Niño conditions. As for the Hadley and Ferrell cells

5  (Fig. S14c), their magnitude increases slightly in both hemispheres, as has been previously stated (Oort and Yienger, 1996).

Walker circulation in the equatorial Pacific is shown in Fig. 7 for CTL and PEN simulations. These are compared to ERA-Interim climatology and ERA-Interim during very strong El Niño conditions respectively. We observe very similar broadscale structures in simulations and the reanalysis. In CTL, air rises in the west Pacific near longitude $150°$ E, flows eastward and sinks around longitude $120°$ W. This is the expected behaviour during normal conditions (Collins et al., 2010) and is also

10  displayed in ERA-Interim climatology. It seems like PlaSim underestimates slightly both upward and downward motions as compared to the reanalysis. In PEN simulation convection is displaced towards the central Pacific, while subsidence in the east is decreased. This is very similar to what happens in ERA-Interim for very strong El Niño events (mean of 1997–1998 and

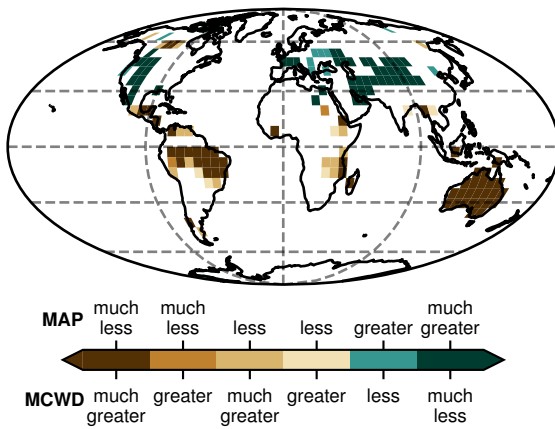

**Figure 9.** Map of gridpoints where the PEN scenario had the largest differences in terms of both MAP and MCWD, with respect to the CTL simulation. Grid lines are spaced every 30° from the Equator and 90° from Greenwich, in latitude and longitude respectively.

2015–2016), although in the reanalysis subsidence in the eastern Pacific ceases almost completely. Overall, PlaSim is able to adequately show the weakening of the Walker circulation during El Niño-like conditions.

### 4.3   Climatological water deficit

Results for both MAP and MCWD are shown in Fig. 8. In terms of MAP, the PEN simulation clearly shows rainfall decline
in the tropics, whereas it increases in northern temperate latitudes. A similar situation is observed for MCWD. As described
in Section 3.3, we combined the differences in MAP and MCWD between scenarios in a single map that is shown in Fig. 9.
This map resembles others that display the global effects of El Niño (e.g. Holmgren et al., 2001). According to these results,
tropical ecosystems seem to be the most vulnerable to a PEN scenario, whereas northern temperate ecosystems could enhance
their primary productivity due to greater water availability. We identify the following regions as the most vulnerable: Central
America, northern and central South America, south-eastern Africa and Australia. On the other hand, some parts of western
North America and a wide band in Eurasia have greater precipitation and cumulative water deficit has diminished. Figure 10
shows how conditions are different in CTL and PEN scenarios, in terms of both MAP and MCWD, in each of these world
regions. In this figure, for every gridpoint in a region of interest in Fig. 9, markers are placed at their PEN value and a solid
line shows the distance to what they are in the CTL scenario. Most gridpoints in the vulnerable regions (Fig. 10b, c, e, f) shift
towards much less precipitation conditions with greater water deficits (see the shift to the left). Using a similar approach as
Malhi et al. (2009) for Amazonia, but in a global context near the tropics, these results indicate that a PEN scenario can shift
terrestrial ecosystems towards more arid conditions. In the case of ecosystems in North America and Eurasia (Fig. 10a, d), a
PEN scenario drives them towards wetter conditions (see the shift to the right).

## 4.4 Warm Pliocene period

The possibility that sustained El Niño conditions might have occurred during the Pliocene Epoch compels us to compare the PEN experiment with previous simulations of the warm Pliocene period. Haywood and Valdes (2004) simulated this period (using a $CO_2$ concentration of $400\,\mathrm{ppm}$) and found global surface temperature to rise. Similarly, near surface temperature
increases in PEN scenario with respect to CTL. In both cases, global warming is greater on land than on the oceans. Regarding average precipitation rate, it increases slightly both in PEN and in the Pliocene simulation with respect to their control scenarios. Another interesting correspondence, is the fact that in Haywood and Valdes (2004) and in PEN, annual mean precipitation decreases in Central America and South America (tropical regions), whereas it increases in the Northern Hemisphere. Particularly for Amazonia, both studies show that average temperature rises and precipitation decreases. Fedorov et al. (2013) report
that in the mid-Pliocene warm period there might have been strengthening of the meridional circulation, leading to aridification of Africa and Australia, which is similar to what we observe in PEN results. However, they also report possible aridification of North America, and this is the opposite of what is seen in PEN (Fig. 9). In spite of the similarities between PEN and the mid-Pliocene warm period simulations, a biomes reconstruction of the Late Pliocene period (Salzmann et al., 2011) seems to disagree with our results, specially for Africa and Australia, since it shows more vegetated regions than we would expect with
the water deficit changes observed in PEN.

## 5   Discussion

Global terrestrial ecosystems can have alternative stable equilibrium sates (Scheffer et al., 2001), and can shift between them when external perturbations and/or internal dynamics force them past critical thresholds (Barnosky et al., 2012; Andersen et al., 2009). Alternative equilibria have already been identified for some biomes in the world. In tropical South America, in
the eastern Amazon river basin, where there is currently a rainforest, simulation results suggest the presence of an alternative savannah (Cerrado) state (Oyama and Nobre, 2003). In both the Sahara and Sahel regions in northern Africa, geological records and modelling results suggest that environmental conditions could have supported a "green Sahara" and a "wet Sahel", but climate shifts and atmosphere–vegetation feedback mechanisms could have stabilized their current contrasting states (Brovkin et al., 1998; Foley et al., 2003). Similarly for the wet tropics in Australia, bi-stability between rainforest and savannah-like
sclerophyll land cover has also been studied (Warman and Moles, 2009). Forest, savannah and desert seem to be three possible alternative stable states for terrestrial ecosystems (Hirota et al., 2011; Staver et al., 2011).

Malhi et al. (2009) studied the possibility of a climate-change-induced state transition in the Amazon rainforest. Estimating water availability and accumulated water stress along a dry season, they identified transition areas between rainforest and savannah alternative states for this large-scale ecosystem. Here we extended this approach to other global ecosystems in order
to evaluate whether a PEN scenario could trigger state transitions not only in Amazonia but also in other global ecosystems. The most noticeable effects suggested by the PlaSim model include a potential greening of western North America, dieback of the Amazon rainforest, drying of south-eastern Africa, and further aridification of Australia.

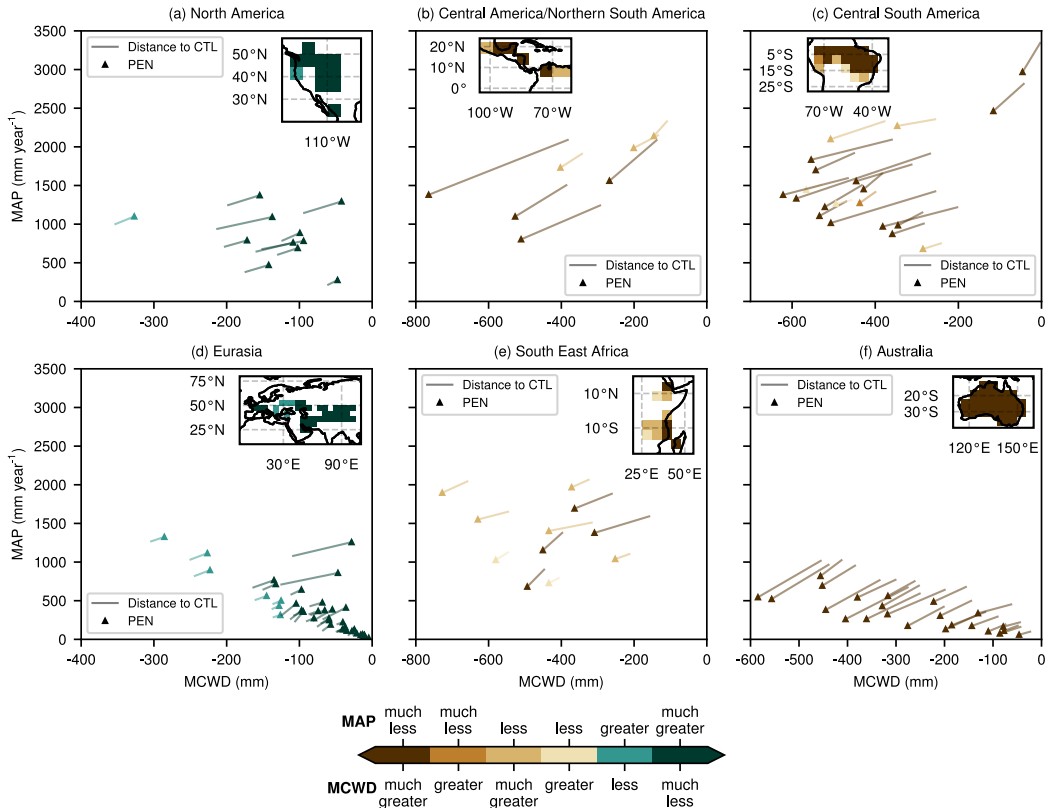

**Figure 10.** Changes in MAP and MCWD in the PEN scenario compared to CTL for different world regions. For every gridpoint shown in the zoomed map axes, we use markers (triangles) to show the value of both variables in the PEN scenario, and a straight line shows the distance to the value they have in the CTL scenario.

Increased precipitation in parts of North America is a known effect of strong El Niño events (Ropelewski and Halpert, 1986) and can be seen in PEN results, as well as reductions in water deficit. In Fig. 10a we observe that significant changes in these variables (with respect to CTL) were greater in gridpoints around the Great Basin region, which is an endorheic watershed that includes arid and semi-arid biomes, and is located close to the Mojave and Sonoran deserts. Several stable states for vegetation in said ecosystems include shrub-lands, short-grass steppes and desert grasslands (Laycock, 1991). Recently it has been found that El Niño is linked to increased primary production in North America (Hu et al., 2019), as well as to growth and persistence of vegetation in these desert areas (Huxman and Smith, 2001). Thus, in a PEN scenario, greater water availability could help avoid further aridification in this region and even trigger greening of some parts of it.

The opposite situation occurs in Central America and northern South America as seen in Fig. 10b. In this case there is less annual precipitation and the water deficit increases. Such results agree with observations during El Niño events which show negative anomalies in several hydrological variables like rainfall and river flows for tropical America (Poveda et al., 2006). In spite of many factors being at play, including the meridional displacement of the ITCZ, complex topography and two different

neighbouring oceans, ENSO has been found to be the greatest cause of inter-annual variability in this region (Poveda et al., 2006) and, therefore, sustained El Niño conditions are very likely to have a noticeable impact as depicted in Fig. 10b. Since this region is considered a biodiversity hotspot in the world (Myers et al., 2000), ecosystem degradation could be related to a demise in its number of biological species due to the drastic changes in water availability observed in PEN. Moreover, hydropower

generation is a common practice in this part of the world (Karmalkar et al., 2011; Poveda et al., 2003) and a change of over 10 % in annual precipitation could hinder energy production.

     Our results show that PEN can be a driver of forest dieback in the Amazon (Fig. 10c). El Niño reduces annual precipitation and increases the water deficit across the dry season for this region (Cox et al., 2004; Malhi et al., 2009). This rainfall reduction could perhaps threaten the existence of parts of the rainforest because resilience to dry conditions is limited and a long-lasting

drought can increase tree mortality (Phillips et al., 2009). Most forest species operate within a narrow margin of hydraulic safety and therefore are vulnerable to drought-induced embolisms which reduce productivity and increase the risk of forest decline (Choat et al., 2012). Additionally, El Niño-induced fires (not simulated by PlaSim) could further exacerbate forest degradation or transition from evergreen to seasonal tropical forests (Barlow and Peres, 2004; Brando et al., 2014). Consequences of this are manifold, but one important aspect to consider is the global impact the Amazon rainforest has as a carbon sink (Cox et al.,

2004; Pan et al., 2011). A PEN-induced dieback in the Amazon could *tip* the rainforest from a sink of carbon to a possible source (Davidson et al., 2012). This could in turn amplify the magnitude of extreme hydrological and weather events that occur during El Niño, such as droughts, in a positive-feedback-like manner (Zemp et al., 2017; Salazar et al., 2018), as well as disrupt mechanisms of river flow regulation (Salazar et al., 2018; Mercado-Bettin et al., 2019) and continental precipitation distribution (Molina et al., 2019; Weng et al., 2018). Near-synchronous effects of this sort between different *tipping elements* could give

rise to a "tipping cascade" that could contribute to a planetary-scale state shift (Lenton and Williams, 2013). Despite potential forest collapse in most of the Amazon, our results also agree with Betts et al. (2008) that some areas in western Amazonia (adjacent to the Andes) are less vulnerable and could act as a refuge for biodiversity. Consequences of a PEN scenario in the Amazon rainforest would have global impact, since it is the largest remaining intact forest in the world, which provide globally significant environmental benefits (Watson et al., 2018; Ellison et al., 2017).

In Eurasia (Fig. 10d) water availability rises in the PEN scenario across central Europe, the Middle East around the Caspian Sea and Central Asia. This is a known feature of El Niño which increases precipitation rates in the region (Ineson and Scaife, 2009). Increased primary production is another behaviour observed during El Niño years in Eurasia (Buermann et al., 2003) that is present in PEN. Particularly in Central Asia, a PEN scenario shows wetter conditions near Karakum and Kyzylkum deserts, where it has been found recently that shrub-lands are extremely sensitive to climatic variations and that drought is the

key driver of vegetation loss (Jiang et al., 2017). Therefore we believe that under sustained El Niño conditions, desertification processes in Central Asia could be inhibited or slowed down. Compared to North America, changes are not as pronounced in precipitation and water deficit, hence we believe greening mechanisms are less likely for Eurasia. The fact that both North America and Eurasia exhibit similar results in PEN scenario can be related to warmer winter and spring seasons and a cooler summer season that leads to greater primary productivity in both places, which is observed in temperate forests during El Niño

years (Malhi et al., 1999).

The influence of ENSO on rainfall patterns in south-eastern Africa was identified long ago (Nicholson and Kim, 1997). This happens directly via atmospheric teleconnections (Ratnam et al., 2014) and also through warming of the Indian and Atlantic oceans (Giannini et al., 2008). El Niño in this region is linked to below average rainfall and droughts (Rouault and Richard, 2005). This is present in our results south of the Great Horn of Africa, where the PEN scenario creates a greater water deficit and much less precipitation (Fig. 10e). Our results agree with Gizaw and Gan (2017), who discussed the impact on Africa of increased frequency of El Niño episodes using climate projections and state-of-the-art general circulation models. They also found south-eastern Africa to be particularly vulnerable. Earlier modelling studies found that such savannah ecosystems in Africa are closely related to climate variability, which keeps the system in between desert-like and forest-like equilibrium states (Zeng and Neelin, 2000). More recent work also associates the savannah equilibrium state to rainfall variability (Staver et al., 2011). Hence, tipping the ENSO permanently could trigger land cover transitions from forest to savannah and/or savannah to desert-like biomes. This is problematic not only in terms of ecosystem degradation, but also in terms of putting at risk food security and hydro-power generation (Lyon, 2014). In the case of West Africa, the impacts of ENSO have been contested because of the many other climatic drivers that come together in this area (Nicholson, 2013). However, ENSO events have been linked to climatic variability in both the Sahel and Guinea coastal regions (Sheen et al., 2017; Joly and Voldoire, 2009; Tippett and Giannini, 2006). These two regions are connected via the West African monsoon in boreal summer, which brings moisture inland from the Atlantic Ocean (Nicholson et al., 2017). During El Niño conditions, a coherent drying pattern for the Sahel has been identified (Pomposi et al., 2016). PlaSim simulation results also support the role of the warm ENSO phase in modulating rainfall in West Africa, and show it leads to a drier state in southern Sahel and Guinea Coast regions (Fig. 8c). This southward expansion of the Sahel dry conditions into Guinea Coast, agrees with the notion of potential enlargement of the world's deserts due to climate change (Zeng and Yoon, 2009). Though this region only has a single gridpoint highlighted in Fig. 9 (due to small changes in water deficit), we believe a PEN scenario could also push its arid and semi-arid ecosystems into vegetation–atmosphere positive feedback mechanisms of desertification.

El Niño events have had different effects on rainfall in Australia, depending on how far east the Walker circulation moves (Wang and Hendon, 2007). In this case, we show that a PEN based on the strong El Niño of 2015–2016 is able to alter climate with potentially severe ecological consequences (Fig. 10f). Likewise, in spite of observed weakening of teleconnections between Australia rainfall and ENSO (Ashcroft et al., 2016), our results indicate that a PEN scenario could cause significant widespread aridification of Australia. Coupling this process with land cover changes, could give rise to synergistic effects that could potentially *tip* Australia's climate towards a much drought-prone state (McAlpine et al., 2009). Furthermore, extreme weather events (such as droughts) and changes in local hydrology, have been identified as key drivers of terrestrial and marine Australian ecosystems towards tipping points (Laurance et al., 2011). Hence, following our results, a PEN could originate a "tipping cascade" in this continent.

Thus the response of tropical and temperate ecosystems to PEN seems to be asymmetric: while tropical ecosystems are degraded (GPP is generally reduced), temperate ecosystems are enhanced (GPP increases). This, combined with land cover changes (human or PEN induced) could turn tropical biomes into global net sources of carbon that, according to current estimates (Pan et al., 2011), could not be counter-balanced by extra-tropical forests.

## 6   Conclusions

This study has provided additional evidence that PlaSim can realistically represent the current state of the Earth's climate system. Despite its coarse resolution and simplified parameterizations, the model is able to display the large-scale patterns of atmospheric dynamics seen in observational data sets. In describing some of its limitations, we found that it displays common biases seen in other state-of-the-art climate models. Therefore, PlaSim provides an adequate framework to study the dynamics of the earth system (as a whole), including complex climate feedback mechanisms, using long simulations.

Our key finding is that a permanent tipping of the ENSO into a very strong warm phase (El Niño) can trigger state transitions in global terrestrial ecosystems, with asymmetric effects between the tropics and extra-tropical regions. We identified some regions that seem to be more susceptible to shifting into more arid biomes: Central America, the Amazon rainforest, south-eastern Africa and Australia. Whereas parts of North America and Eurasia could become more productive. These results also suggest the possibility that a permanent El Niño can trigger tipping cascades. It is important to mention that despite there being other sources of variability for these ecosystems (for instance the Atlantic Ocean temperatures variation), here we focused only on the consequences that an observed-very-strong-El Niño-based SST climatology could have on continental areas, without linking any simulated anomalies to a specific region in space. This means that despite the Pacific Ocean shows the largest anomalies (as in El Niño) we cannot attribute all of our results to changes only in this region. Even though a modelling study of this sort is not enough to prove that such state shifts will occur in a climate change scenario, it does show a consistent picture, and should raise concern about the conservation of global ecosystems and related water resources management practices in a world with more frequent and/or intense El Niño events (Latif et al., 2015; Cai et al., 2014).

*Data availability.*   All observational data sets and reanalyses used in this study are publicly available online. Model results are readily available upon request.

*Author contributions.*   MD-V performed the simulations. All authors participated in the design of the experiments, the discussion of the results, and the writing of the paper.

*Competing interests.*   The authors declare that they have no conflict of interest.

*Acknowledgements.*   MD-V was funded by Universidad de Antioquia through the "Estudiante Instructor" program. JFS and AMR were partially supported by "Programa de investigación en la gestión de riesgo asociado con cambio climático y ambiental en cuencas hidrográficas" (UT-GRA), Convocatoria 543–2011 Colciencias. We thank Daniel Ruiz Carrascal and Luis F. Salazar for insightful comments on an early

version of this manuscript. We also thank Stephen Sitch and Lina Mercado for their valuable remarks. Finally, we thank Rui A. P. Perdigão and two anonymous reviewers for their helpful observations.

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
