# Peer review of "Tipping the ENSO into a permanent El Niño can trigger state transitions in global terrestrial ecosystems"

_Earth System Dynamics, 2019_

## Referee Comment (RC1) · Anonymous Referee #1 · 3 Jun 2019

Recommendation: Accepted after major revisions

The authors present a thorough study in an intermediate complex global model (PlaSim) about the impact that a permanent ENSO state might have in the global energy and water balances and also in the global atmospheric circulation and terrestrial ecosystems. The results in the manuscript and in the supplementary material are generally very well presented. However, there are some main issues that must be revised.

1 - Despite the utility of this kind of simulations, there is no proof that a permanent ENSO could be a climatic equilibrium state, consistent with the present CO2 concentrations. Both in CTL and PEN simulations, CO2 is kept constant at 360 ppm, near the

global mean value of 1995, whereas the present (2018-19) mean value is around 416 ppm. Authors agree that ENSO is becoming reinforced under a larger radiative ($CO_2$) forcing. Therefore, why have not the authors set the CO2 concentrations to 2018 values. The fact that CO2 is kept at the 1995 values introduces an inconsistency. Authors shall present the impact (at least in the water deficit) of changing simultaneously SST and CO2 forcing. There is evidence that some Permanent El Niño-Like Conditions have occurred During the Pliocene Warm Period (Wara et al. 2005). Authors shall compare the forcing of PEN simulations with those of Pliocene.

2 – The model bias given by the difference CTL minus observations is of the same order of that of some GCMs. However, the mean difference between PEN and CTL simulations is much lower in amplitude for certain fields (e.g. land surface temperature and precipitation) and locations, thus raising duties about the significance of the ENSO impact. Conclusions are only valid under the hypothesis that model bias is the same, both in climatological SST conditions and ENSO conditions. The confidence in the impact results is only valid by assessing the model bias under ENSO conditions. It is quite simple to do so through the difference PEN minus composite mean observations under ENSO (by choosing the set of larger ENSO events).

3 – Authors refer the sensitivity of model climatology to initial conditions by considering 3 perturbed initial condition fields. However, the results of that sensitivity and the t-Student tests are not explicitly presented.

4 – Quantitative diagnostics of the Walker circulation (bias and impact) shall be included.

Reference

Wara, MW.; Ravelo, AC., Delaney, ML. 2005. Permanent El Niño-Like Conditions During the Pliocene Warm Period. Science 29 Jul 2005: Vol. 309, Issue 5735, pp. 758-761 ,DOI: 10.1126/science.1112596

---

## Referee Comment (RC2) · Anonymous Referee #2 · 12 Jul 2019

Overall, the manuscript is well written, aptly detailing a thorough and comprehensive investigation on a highly relevant topic to ESD. Technically, the study is well developed and discussed, making for a smooth and instructive read.

This being said, my recommendations would be to make it further clear to the readers that this study has exploratory nature, portraying a modelling exercise based on assumptions that may or may not necessarily correspond to the reality. In this sense, I agree with comments 1 and 2 of the other reviewer.

The more detailed discussion of working assumptions e.g. on stability and equilibria and how realistic or not they are would thus further benefit the realistic framing of

the simulation results in a way that would allow the reader to better appreciate their relevance and value.

I would also appreciate if the authors could provide thorougher sensitivity tests to perturbations in initial conditions in order to further substantiate the relatively basic sensitivity assessment that, while informative, could strongly benefit from such tests.

In conclusion, I am mostly satisfied with the manuscript quality. The raised concerns are relatively minor. Therefore, I would definitely recommend publication after these concerns are addressed.

---

## Author Comment (AC2) · 15 Jul 2019

Comment:

*Overall, the manuscript is well written, aptly detailing a thorough and comprehensive investigation on a highly relevant topic to ESD. Technically, the study is well developed and discussed, making for a smooth and instructive read.*

*This being said, my recommendations would be to make it further clear to the readers that this study has exploratory nature, portraying a modelling exercise based on assumptions that may or may not necessarily correspond to the reality. In this sense, I*

[Figure]

*agree with comments 1 and 2 of the other reviewer.*

Response:

We thank the reviewer for his/her comments. We will add a short sentence near the end of the introduction section that will emphasize the exploratory nature of our paper. Please also refer to our response to comments 1 and 2 by Anonymous Referee #1. Also please refer to the supplementary material to Referee Comment #1 where you will find new figures supporting our responses, which numbering is prefixed by a 'C'.

Comment:

*The more detailed discussion of working assumptions e.g. on stability and equilibria and how realistic or not they are would thus further benefit the realistic framing of the simulation results in a way that would allow the reader to better appreciate their relevance and value. I would also appreciate if the authors could provide thorougher sensitivity tests to perturbations in initial conditions in order to further substantiate the relatively basic sensitivity assessment that, while informative, could strongly benefit from such tests.*

Response:

The revised manuscript will include extended results and discussion subsections to further explain our assumptions and results, as well as their implications. This includes clarification of the sensitivity tests to perturbation in initial conditions (include Fig. C7 as part of the manuscript's supplementary material).

Comment:

*In conclusion, I am mostly satisfied with the manuscript quality. The raised concerns are relatively minor. Therefore, I would definitely recommend publication after these concerns are addressed.*

Response:

Thank you for your recommendation.

---

## Author Comment (AC3) · 15 Jul 2019

Comment:

*Recommendation: Accepted after major revisions*

*The authors present a thorough study in an intermediate complex global model (PlaSim) about the impact that a permanent ENSO state might have in the global energy and water balances and also in the global atmospheric circulation and terrestrial ecosystems. The results in the manuscript and in the supplementary material are generally very well presented. However, there are some main issues that must be revised.*

[Figure]

Response:

We thank the reviewer for his/her comments. Specific answers to each comment are provided below. All support figures for our comments can be found as supplementary material to this comment, with prefix 'C' as part of the numbering

Comment:

*1 - Despite the utility of this kind of simulations, there is no proof that a permanent ENSO could be a climatic equilibrium state, consistent with the present $CO_2$ concentrations. Both in CTL and PEN simulations, $CO_2$ is kept constant at 360 ppm, near the global mean value of 1995, whereas the present (2018-19) mean value is around 416 ppm. Authors agree that ENSO is becoming reinforced under a larger radiative ($CO_2$) forcing. Therefore, why have not the authors set the $CO_2$ concentrations to 2018 values. The fact that $CO_2$ is kept at the 1995 values introduces an inconsistency. Authors shall present the impact (at least in the water deficit) of changing simultaneously SST and $CO_2$ forcing. There is evidence that some Permanent El Niño-Like Conditions have occurred During the Pliocene Warm Period (Wara et al. 2005). Authors shall compare the forcing of PEN simulations with those of Pliocene.*

Response:

Two things:

1.1 - You are right. We update all results using a $CO_2$ value of 415 ppm, which, as you pointed out, is closer to current estimates of this variable (Dlugokencky and Tans, 2019). This change directly affects the long-wave radiation and dynamic vegetation modules in the model. However, differences among simulations are modest enough (Fig. C1 and C2) that we only deem it necessary to present the results obtained with this new 415 ppm $CO_2$ value. In terms of the water deficit, the new results show many of the same patterns previously discussed and only some minor changes in the Northern Hemisphere can be seen (Fig. C3). We will extend our discussion to include these

minor changes. For the manuscript, this means all figures except those that pertain the methodology (Fig. 1 and Fig. S1) will be updated. Experimental setup section will be updated accordingly.

1.2 - Wara et al. (2005) suggest the possibility that permanent El Niño-like conditions may have occurred during the warm early Pliocene period. Using environmental reconstructions from isotopes and bioindicators, they found that the zonal west-to-east gradient of sea surface temperature (SST) was very similar to the one observed during modern El Niño events. Although more recent studies suggest that their reconstruction may underestimate the Pacific warm pool temperature and its variability (Zhang et al., 2014), it is still interesting to compare PEN scenario's results with those of Pliocene simulations. In Fig. C4 we compare the SST boundary conditions of the PEN scenario with the paleoenvironmental reconstruction data set PRISM3 (Dowsett et al., 2009), which covers the mid-Pliocene warm period and has been used in many Pliocene modeling experiments (Haywood et al., 2016). PEN forcing was on average about 1°C warmer in the tropics, though the differences in this region are not all significant. PEN was cooler than PRISM3 elsewhere. The west-to-east zonal gradient was computed for both forcings, as well as that of the CTL simulation, at the sites shown in Fig. C4. For PEN the gradient was 1.5°C, whereas for PRISM3 it was 1.8°C. CTL simulation had a gradient of 3.1°C. Since this gradient is a good indicator of the strength of the Walker circulation (Wara et al., 2005), we can say that in the PEN scenario it was weaker than in the mid-Pliocene warm period, and much weaker than in the CTL scenario. Haywood and Valdes (2004) simulated the mid-Pliocene warm period (using a $CO_2$ concentration of 400 ppm) and found global surface temperature to rise. Near surface temperature also rose in PEN scenario with respect to its control scenario. Also in both studies warming was greater on land than on the oceans. Regarding precipitation rate, it also increases slightly in PEN and in the Pliocene simulation with respect to their control scenarios. Another resemblance is the fact that both in Haywood and Valdes (2004) and in PEN, annual mean precipitation decreases in Central and South America, whereas it increases in the Northern Hemisphere. Particularly for the Amazon region, both studies show that temperatures rise and precipitation decreases. Fedorov et al. (2011) report that in the mid-Pliocene warm period there might have been strengthening of the meridional circulation leading to aridification of Africa, Australia and North America, which is very similar to what we observe in PEN results. In spite of the similarities between PEN and the mid-Pliocene warm period, a biomes reconstruction of the Late Pliocene period (Salzmann et al., 2011) seems to disagree with our results, specially for Africa and Australia, since it shows more vegetated regions than we would expect with the water deficit changes observed in PEN. We believe PEN to be a more arid equilibrium partly because land cover in this scenario (which is similar to current global land cover) has a smaller extent of vegetation than that of the Pliocene. This will be included as a Pliocene-related subsection in the discussion section.

Comment:

*2 - The model bias given by the difference CTL minus observations is of the same order of that of some GCMs. However, the mean difference between PEN and CTL simulations is much lower in amplitude for certain fields (e.g. land surface temperature and precipitation) and locations, thus raising duties about the significance of the ENSO impact. Conclusions are only valid under the hypothesis that model bias is the same, both in climatological SST conditions and ENSO conditions. The confidence in the impact results is only valid by assessing the model bias under ENSO conditions. It is quite simple to do so through the difference PEN minus composite mean observations under ENSO (by choosing the set of larger ENSO events).*

Response:

2 - Since our original results changed slightly, due to setting atmospheric $CO_2$ to 415 ppm, we first checked that CTL biases were still in the same order of that of other general circulation models. The new CTL biases were very similar to those shown in Fig. 3 of the unrevised manuscript. Then we followed your recommendation and plotted Fig. C5, in which we compare PEN simulation versus observational databases

during strong El Niño events. MODIS is an exception since data only covers period 2000 - 2015, so it does not include any strong El Niño events, but only moderate ones. We again find similarities between PEN and observations in the broad scale patterns of these variables. Bias panels for PEN in Fig. C5 are very similar to those of CTL in Fig. 3 in the unrevised manuscript. For ease of comparison we have plotted Fig. C6 in which all these bias panels are shown, as well as their absolutes difference. Overall, we see that model bias is similar for climatological and El Niño conditions. Only for MODIS we see relatively large bias differences, most likely because the database does not include any strong El Niño events. Hence we are confident that differences between scenarios are a realistic representation of the effects caused by the strong El Niño SST forcing. This will be included in the results section.

Comment:

*3 - Authors refer the sensitivity of model climatology to initial conditions by considering perturbed initial condition fields. However, the results of that sensitivity and the t-Student tests are not explicitly presented.*

Response:

3 - Although the model does not seem to display great sensitivity to initial conditions, we believe that it is an important part of atmospheric modeling experiments to test for chaos and this is why we have ensemble mean results rather than single simulation results. As shown in Fig. C7 for the CTL simulation mean values, differences among ensemble members are nonetheless very small. Notice that in Fig. C7 we have used a significance level of 0.1 to be able to at least show that they are in fact different simulations. When we used 0.05 instead, most maps showed up empty. This will be clarified in the experimental setup section.

Comment:

*4 - Quantitative diagnostics of the Walker circulation (bias and impact) shall be in-*

[Figure]

*cluded.*

Response:

4 - Walker circulation in the equatorial Pacific is now shown in Fig. C8 for both CTL and PEN simulations. In both PlaSim simulations we see that the pattern is not as coherent as displayed in ERA-Interim, however the same broad scale structures are observed. In CTL, air rises in the west Pacific near longitude 150°E, flows eastward and sinks around longitude 120°W. This the expected behaviour during normal conditions (Collins et al., 2010) and is also observed in ERA-Interim climatology. It seems like PlaSim underestimates both upward and downward motions as compared to the reanalysis. In PEN simulation we see convection is displaced into the central Pacific while subsidence in the east is decreased. This is very similar to what happens in ERA-Interim for strong El Niño years (mean of 1997-98 and 2015-16), although in the reanalysis subsidence in the east Pacific ceases almost completely. In this case we also see PlaSim underestimating vertical motions. Overall, PlaSim is able to adequately show the weakening of the Walker circulation during El Niño-like conditions. This will be included in the results section.

[Figure]

**References**

Collins, M., An, S. I., Cai, W., Ganachaud, A., Guilyardi, E., Jin, F. F., ...   Vecchi, G. (2010).  The impact of global warming on the tropical Pacific Ocean and El Niño. Nature Geoscience, 3(6), 391.

Dlugokencky, E. and Tans, P.: Globally averaged marine surface monthly mean data, www.esrl.noaa.gov/gmd/ccgg/trends/, 2019.

Dowsett, H. J., Robinson, M. M.,  Foley, K. M. (2009).  Pliocene three-dimensional global ocean temperature reconstruction. Climate of the Past, 5(4), 769-783.

Haywood, A. M., Dowsett, H. J.,  Dolan, A. M. (2016). Integrating geological archives and climate models for the mid-Pliocene warm period.  Nature communications, 7, 10646.

Salzmann, U., Williams, M., Haywood, A. M., Johnson, A. L., Kender, S.,  Zalasiewicz, J. (2011).  Climate and environment of a Pliocene warm world.  Palaeogeography, Palaeoclimatology, Palaeoecology, 309(1-2), 1-8.

Wara, M. W., Ravelo, A. C.,  Delaney, M. L. (2005). Permanent El Niño-like conditions during the Pliocene warm period. Science, 309(5735), 758-761.

Zhang, Y. G., Pagani, M.,  Liu, Z. (2014).  A 12-million-year temperature history of the tropical Pacific Ocean. Science, 344(6179), 84-87.

Please also note the supplement to this comment:
https://www.earth-syst-dynam-discuss.net/esd-2019-14/esd-2019-14-AC3-supplement.pdf

[Figure]

**Supplement:**

**Supplementary information for Referee Comment #1 on "*Tipping the ENSO into a permanent El Niño can trigger state transitions in global terrestrial ecosystems*"**

Mateo Duque-Villegas[1], Juan F. Salazar[1], and Angela M. Rendón[1]

[1]GIGA, Escuela Ambiental, Facultad de Ingeniería, Universidad de Antioquia, Medellín, Colombia

**Correspondence:** Juan F. Salazar (juan.salazar@udea.edu.co)

**List of Figures**

[Figure]

**Figure C1.** Differences among PEN simulations with different atmospheric $CO_2$ values for variables: (a) near surface temperature, (b) total precipitation rate, (c) gross primary production and (d) mean sea-level surface pressure. $CO_2$ values are shown as subscripts. White is for statistically non-significant differences ($\alpha = 0.05$). Gridlines are spaced every $30°$ in the parallels from the Equator, and every $90°$ in the meridians from Greenwich. This figure will not be included in the revised manuscript, but perhaps as supplementary material.

[Figure]

**Figure C2.** Differences among PEN simulations with different atmospheric $CO_2$ values for variables: (a) air temperature, (b) zonal wind speed and (c) zonal mean meridional mass streamfunction. White is for statistically non-significant differences ($\alpha = 0.05$). This figure will not be included in the revised manuscript, but perhaps as supplementary material.

[Figure]

**Figure C3.** Comparison in water deficit results between simulations with different atmospheric $CO_2$ values: (a) 360 ppm and (b) 415 ppm. The meaning of colors is the same as explained in the manuscript. Gridlines are spaced every $30°$ in the parallels from the Equator, and every $90°$ in the meridians from Greenwich. This figure will not be included in the revised manuscript, but perhaps as supplementary material.

[Figure]

**Figure C4.** (top) Sea surface temperature forcing in (a) PEN simulation, (b) PRISM3 data set and (c) their differences. Gridlines are spaced every 30° in the parallels from the Equator, and every 90° in the meridians from Greenwich. (bottom) Equatorial Pacific sea surface temperature forcing in (d) PEN simulation, (e) PRISM3 data set and (f) their differences. Markers show the west (158° E, 2.8° N) and east (96° W, 2.8° N) sites used to compute the zonal gradient. In all bias panels white is for statistically non-significant differences ($\alpha = 0.05$). This figure will not be included in the revised manuscript, but perhaps as supplementary material.

[Figure]

**Figure C5.** Annual mean results for PEN simulation compared with observational data for El Niño years: near surface temperature in (a) PEN, (b) HadCRUT4 mean of 1997 – 98 and 2015 – 16, and (c) PEN bias; total precipitation rate in (d) PEN, (e) GPGP mean of 1997 – 98 and 2015 – 16, and (f) PEN bias; gross primary production in (g) PEN, (h) MODIS mean of 2002 – 03 and 2009 – 10 and (i) PEN bias; and mean sea-level surface pressure in (j) PEN, (k) HadSLP2 mean of 1982 – 83 and 1997 – 98. Gridlines are spaced every 30° in the parallels from the Equator, and every 90° in the meridians from Greenwich. In all bias panels white is for statistically non-significant differences ($\alpha$ = 0.05). This figure will not be included in the revised manuscript, but perhaps as supplementary material.

[Figure]

**Figure C6.** Bias panels from CTL simulation compared with PEN and their absolutes differences for variables: near surface temperature bias in (a) CTL, (b) PEN and (c) their differences; total precipitation rate bias in (d) CTL, (e) PEN and (f) their differences; gross primary production bias in (g) CTL, (h) PEN and (i) their differences; and mean sea-level surface pressure bias in (j) CTL, (k) PEN and (l) their differences. Gridlines are spaced every $30°$ in the parallels from the Equator, and every $90°$ in the meridians from Greenwich. In all panels white is for statistically non-significant differences ($\alpha = 0.05$). This figure will not be included in the revised manuscript, but perhaps as supplementary material.

[Figure]

**Figure C7.** Annual mean differences of ensemble members versus the ensemble mean for CTL simulation for variables: near surface temperature (a, b, c), total precipitation rate (d, e, f), gross primary production (g, h, i) and mean sea-level surface pressure (j, k, l). Gridlines are spaced every $30°$ in the parallels from the Equator, and every $90°$ in the meridians from Greenwich. In all panels white is for statistically non-significant differences ($\alpha = 0.1$). This figure will not be included in the revised manuscript, but perhaps as supplementary material.

[Figure]

**Figure C8.** Meridional divergent circulation in the equatorial Pacific for (a) CTL simulation, (b) climatological ERA-Interim and (c) CTL bias in vertical velocity values; as well as for (d) PEN simulation, (e) ERA-Interim mean of 1997 – 98 and 2015 – 16, and (f) PEN bias in vertical velocity values. Vectors are plotted using the divergent meridional wind component and the negative vertical velocity in pressure units ($-\omega$), averaged over latitudes 5° S to 5° N. Filled contours show the magnitude and value of the vertical velocity. In all bias panels white is for statistically non-significant differences ($\alpha = 0.05$). This figure will be included in the revised manuscript.

---

## Author Response (AR1)

**Authors' response to the reviews**

Mateo Duque-Villegas[1], Juan F. Salazar[1], and Angela M. Rendón[1]

[1]GIGA, Escuela Ambiental, Facultad de Ingeniería, Universidad de Antioquia, Medellín, Colombia

We thank the handling editor and the anonymous reviewers for their time and effort spent in reviewing our manuscript. We believe that this reviewing process has improved our original submission. Here we provide a point-by-point response to the reviews. Comments will be shown in *italics* and our response immediately below in normal typeface. Figures with prefix "C" should be looked for in the supplementary material to the reviewers comments (https://www.earth-syst-dynam-discuss. net/esd-2019-14/esd-2019-14-AC3-supplement.pdf). Some of these figures are now part of the supplementary material of the revised manuscript.

**1 Handling Editor Comments**

**Comment**

*Dear Authors*

*Mateo, Juan and Angela,*

*Thank you very much for submitting your manuscript to the consideration of Earth System Dynamics. Having carefully conducted a preliminary assessment, I deem it relevant to the interdisciplinary scope of the journal and suitable for Review and Discussion at ESD-Discussions.*

*With very best wishes,*

*Rui Perdigão*

*(ESD Editor)*

*Side Note: The geospatial plots have high graphical quality and most have very good colour schemes (the red-beige-blue being particularly safe and elegant). In general, I would simply suggest avoiding the red-green schemes since it has been brought to my attention that a small but significant part of the population (and of our readership) has red-green colour blindness. This can be addressed at a later stage e.g. during revisions or typesetting depending on the outcome of the review process.*

**Response**

Thank you for your good wishes. Attending to your suggestion, we have updated all of our figures to not include any red-green colour shemes.

**Comment**

*Dear Authors,*

*Thank you very much for your responses to the referee reports. Having taken the manuscript, the reviews and the author responses into consideration, the work has good potential for fruition, especially with the improvements suggested by the referees, pledged by the authors, and which are hereby agreed with editorially.*

*I look forward to seeing the revised version.*

*Good luck with the revisions.*

*With best regards,*

*Rui Perdigão*

**Response**

Once again thank you for your good wishes.

**2 Anonymous Referee #1 Comments**

**Comment**

*Recommendation: Accepted after major revisions*

*The authors present a thorough study in an intermediate complex global model (PlaSim) about the impact that a permanent ENSO state might have in the global energy and water balances and also in the global atmospheric circulation and terrestrial ecosystems. The results in the manuscript and in the supplementary material are generally very well presented. However, there are some main issues that must be revised.*

**Response**

We thank the reviewer for his/her comments. Specific answers to each comment are provided below.

**Comment**

*1 - Despite the utility of this kind of simulations, there is no proof that a permanent ENSO could be a climatic equilibrium state, consistent with the present $CO_2$ concentrations. Both in CTL and PEN simulations, $CO_2$ is kept constant at 360 ppm, near the global mean value of 1995, whereas the present (2018-19) mean value is around 416 ppm. Authors agree that ENSO is becoming reinforced under a larger radiative ($CO_2$) forcing. Therefore, why have not the authors set the $CO_2$ concentrations to 2018 values. The fact that $CO_2$ is kept at the 1995 values introduces an inconsistency. Authors shall present the impact (at least in the water deficit) of changing simultaneously SST and $CO_2$ forcing. There is evidence that some Permanent El Niño-Like Conditions have occurred During the Pliocene Warm Period (Wara et al., 2005). Authors shall compare the forcing of PEN simulations with those of Pliocene.*

**Response**

Two things:

1.1 - You are right. We have updated all results using a $CO_2$ value of $415\,\mathrm{ppm}$, which, as you pointed out, is closer to current estimates of this variable (Dlugokencky and Tans, 2019). This change directly affects the long-wave radiation and dynamic vegetation modules in the model. However, differences among simulations are modest enough (Figs. C1 and C2) that we only deemed it necessary to present the results obtained with this new $415\,\mathrm{ppm}$ $CO_2$ value. In terms of the water deficit, the new

results show many of the same patterns previously discussed and only some minor changes in the Northern Hemisphere can be seen (Fig. C3). We have extended our discussion to include these minor changes. For the manuscript, this meant all figures except those that pertain the methodology (Fig. 1 and Fig. S1) were updated. Experimental setup section was be updated accordingly. Figures C1, C2 and C3 were not included in the manuscript or supplementary material.

1.2 - Wara et al. (2005) suggest the possibility that permanent El Niño-like conditions may have occurred during the warm early Pliocene period. Using environmental reconstructions from isotopes and bioindicators, they found that the zonal west-to-east gradient of sea surface temperature (SST) was very similar to the one observed during modern El Niño events. Although more recent studies suggest that their reconstruction may underestimate the Pacific warm pool temperature and its variability (Zhang et al., 2014), it is still interesting to compare PEN scenario's results with those of Pliocene simulations. In Fig. C4 we compare the SST boundary conditions of the PEN scenario with the paleoenvironmental reconstruction data set PRISM3 (Dowsett et al., 2009), which covers the mid-Pliocene warm period and has been used in many Pliocene modeling experiments (Haywood et al., 2016). PEN forcing was on average about 1° C warmer in the tropics, though the differences in this region are not all significant. PEN was cooler than PRISM3 elsewhere. The west-to-east zonal gradient was computed for both forcings, as well as that of the CTL simulation, at the sites shown in Fig. C4. For PEN the gradient was 1.5° C, whereas for PRISM3 it was 1.8° C. CTL simulation had a gradient of 3.1° C. Since this gradient is a good indicator of the strength of the Walker circulation (Wara et al., 2005), we can say that in the PEN scenario it was weaker than in the mid-Pliocene warm period, and much weaker than in the CTL scenario. Haywood and Valdes (2004) simulated the mid-Pliocene warm period (using a $CO_2$ concentration of $400\,\mathrm{ppm}$) and found global surface temperature to rise. Near surface temperature also rose in PEN scenario with respect to its control scenario. Also in both studies warming was greater on land than on the oceans. Regarding precipitation rate, it also increases slightly in PEN and in the Pliocene simulation with respect to their control scenarios. Another resemblance is the fact that both in Haywood and Valdes (2004) and in PEN, annual mean precipitation decreases in Central and South America, whereas it increases in the Northern Hemisphere. Particularly for the Amazon region, both studies show that temperatures rise and precipitation decreases. Fedorov et al. (2013) report that in the mid-Pliocene warm period there might have been strengthening of the meridional circulation leading to aridification of Africa, Australia and North America, which is very similar to what we observe in PEN results. In spite of the similarities between PEN and the mid-Pliocene warm period, a biomes reconstruction of the Late Pliocene period (Salzmann et al., 2011) seems to disagree with our results, specially for Africa and Australia, since it shows more vegetated regions than we would expect with the water deficit changes observed in PEN. This was included as a Pliocene-related subsection in the discussion section. Figure C4 was added to the supplementary material as Fig. S3.

**Comment**

*2 - The model bias given by the difference CTL minus observations is of the same order of that of some GCMs. However, the mean difference between PEN and CTL simulations is much lower in amplitude for certain fields (e.g. land surface temperature and precipitation) and locations, thus raising duties about the significance of the ENSO impact. Conclusions are only valid under the hypothesis that model bias is the same, both in climatological SST conditions and ENSO conditions. The confidence in the impact results is only valid by assessing the model bias under ENSO conditions. It is quite simple to do so through the difference PEN minus composite mean observations under ENSO (by choosing the set of larger ENSO events).*

**Response**

2 - Since our original results changed slightly, due to setting atmospheric $CO_2$ to $415\,\mathrm{ppm}$, we first checked that CTL biases were still in the same order of that of other general circulation models. The new CTL biases were very similar to those shown in Fig. 3 of the unrevised manuscript. Then we followed your recommendation and plotted Fig. C5, in which we compare PEN simulation versus observational databases during strong El Niño events. MODIS is an exception since data only covers period 2000 - 2015, so it does not include any strong El Niño events, but only moderate ones. We again find similarities between PEN and observations in the broad scale patterns of these variables. Bias panels for PEN in Fig. C5 are very similar to those of CTL in Fig. 3 in the manuscript. For ease of comparison we have plotted Fig. C6 in which all these bias panels are shown, as well as their absolutes difference. Overall, we see that model bias is similar for climatological and El Niño conditions. Only for MODIS we see relatively large bias differences, most likely because the database does not include any strong El Niño events. Hence we are confident that differences between scenarios are a realistic representation of the effects caused by the strong El Niño SST forcing. This was included in the results section. Also Figs. C5 and C6 were included in the supplementary material as Figs. S10 and S11 respectively.

**Comment**

*3 - Authors refer the sensitivity of model climatology to initial conditions by considering perturbed initial condition fields. However, the results of that sensitivity and the t-Student tests are not explicitly presented.*

**Response**

3 - Although the model does not seem to display great sensitivity to initial conditions, we believe that it is an important part of atmospheric modeling experiments to test for chaos and this is why we have ensemble mean results rather than single simulation results. As shown in Fig. C7 for the CTL simulation mean values, differences among ensemble members are nonetheless very small. Notice that in Fig. C7 we have used a significance level of 0.1 to be able to at least show that they are in fact different simulations. When we used 0.05 instead, most maps showed up empty. This was clarified in the experimental setup section. Also Fig. C7 was included in the supplementary material as Fig. S2.

**Comment**

*4 - Quantitative diagnostics of the Walker circulation (bias and impact) shall be included.*

**Response**

4 - Walker circulation in the equatorial Pacific is now shown in Fig. C8 for both CTL and PEN simulations. In both PlaSim simulations we see that the pattern is not as coherent as displayed in ERA-Interim, however the same broad scale structures are observed. In CTL, air rises in the west Pacific near longitude $150°$ E, flows eastward and sinks around longitude $120°$ W. This the expected behaviour during normal conditions (Collins et al., 2010) and is also observed in ERA-Interim climatology. It seems like PlaSim underestimates both upward and downward motions as compared to the reanalysis. In PEN simulation we see convection is displaced into the central Pacific while subsidence in the east is decreased. This is very similar to what happens in ERA-Interim for strong El Niño years (mean of 1997-98 and 2015-16), although in the reanalysis subsidence in the east Pacific ceases almost completely. In this case we also see PlaSim underestimating vertical motions. Overall, PlaSim is

able to adequately show the weakening of the Walker circulation during El Niño-like conditions. This was be included in the results section. Figure C8 was included in the manuscript as Fig. 7.

**3 Anonymous Referee #2 Comments**

**Comment**

*Overall, the manuscript is well written, aptly detailing a thorough and comprehensive investigation on a highly relevant topic to ESD. Technically, the study is well developed and discussed, making for a smooth and instructive read.*
*This being said, my recommendations would be to make it further clear to the readers that this study has exploratory nature, portraying a modelling exercise based on assumptions that may or may not necessarily correspond to the reality. In this sense, I agree with comments 1 and 2 of the other reviewer.*

**Response**

We thank the reviewer for his/her comments. We added a short sentence near the end of the introduction section that emphasizes the exploratory nature of our paper. Please also refer to our response to comments 1 and 2 by Anonymous Referee #1.

**Comment**

*The more detailed discussion of working assumptions e.g. on stability and equilibria and how realistic or not they are would thus further benefit the realistic framing of the simulation results in a way that would allow the reader to better appreciate their relevance and value. I would also appreciate if the authors could provide thorougher sensitivity tests to perturbations in initial conditions in order to further substantiate the relatively basic sensitivity assessment that, while informative, could strongly benefit from such tests.*

**Response**

The revised manuscript includes extended results and discussion subsections to further explain our assumptions and results, as well as their implications. This includes clarification of the sensitivity tests to perturbation in initial conditions. We have included Fig.C7 as Fig. S2 in the supplementary material.

**Comment**

*In conclusion, I am mostly satisfied with the manuscript quality. The raised concerns are relatively minor. Therefore, I would definitely recommend publication after these concerns are addressed.*

**Response**

Thank you for your recommendation.

**List of relevant changes**

Mateo Duque-Villegas[1], Juan F. Salazar[1], and Angela M. Rendón[1]

[1]GIGA, Escuela Ambiental, Facultad de Ingeniería, Universidad de Antioquia, Medellín, Colombia

A major change in our study is that our simulations had to be updated to use the $CO_2$ value of $415\,\mathrm{ppm}$. This meant that all figures in the manuscript had to be updated, as well as the in-text number results. In doing so, we attended the editor's suggestion to keep away from red-green colour schemes. Luckily, changes due to the new atmospheric carbon dioxide were modest enough that our discussion is still supported with the new results. Following the comments by anonymous reviewers, the revised manuscript now includes:

- Updated figures.

- Updated global average numbers.

- A short clarification about the exploratory nature of our study.

- A short clarification about the sensitivity of the model to initial conditions.

- A short clarification about CTL and PEN biases being similar.

- A comparison of PEN scenario's forcing with Pliocene Epoch sea surface temperature data.

- A comparison of PEN scenarios's results with Pliocene Epoch simulations.

- Quantitative results of the Walker Circulation for CTL, PEN and ERA-Interim are shown in new Fig. 7.

- Figure 9 has been updated to better display changes in MAP and MCWD for different world regions. In the revised manuscript it is now Fig. 10.

- Discussion of new regions: North America, Central America and Eurasia, which stand out more in the new results.

Supplementary material was also updated and now includes:

- Tables S3, S4, and S5 summarize all numbers shown in text, and they also contain the results of Student's t-tests for global averages comparisons between scenarios.

- Fig. S2 that shows how ensemble members do not differ much.

- Fig. S3 that supports Pliocene-related discussion.

- Figs. S9 and S10 are now Figs. S4 and S5. This is because they are referenced in text before others.

– Figs. S2-S5 are now Figs. S6-S9.

– Fig. S10 shows comparison of PEN with El Niño observations.

– Fig. S11 shows that biases between CTL and PEN are similar.

– Figs. S6-9 are now Figs. S12-14.

Accompanying this list there is a marked-up version of the manuscript that uses bold typeface to highlight new content.

[revised manuscript text omitted]